# Spotless, a reproducible pipeline for benchmarking cell type deconvolution in spatial transcriptomics

**Chananchida Sang-aram[1,2], Robin Browaeys[1,2], Ruth Seurinck[1,2], Yvan Saeys[1,2]***

[1]Data Mining and Modelling for Biomedicine, VIB Center for Inflammation Research, Ghent, Belgium; [2]Department of Applied Mathematics, Computer Science and Statistics, Ghent University, Ghent, Belgium

**Abstract** Spatial transcriptomics (ST) technologies allow the profiling of the transcriptome of cells while keeping their spatial context. Since most commercial untargeted ST technologies do not yet operate at single-cell resolution, computational methods such as deconvolution are often used to infer the cell type composition of each sequenced spot. We benchmarked 11 deconvolution methods using 63 silver standards, 3 gold standards, and 2 case studies on liver and melanoma tissues. We developed a simulation engine called *synthspot* to generate silver standards from single-cell RNA-sequencing data, while gold standards are generated by pooling single cells from targeted ST data. We evaluated methods based on their performance, stability across different reference datasets, and scalability. We found that cell2location and RCTD are the top-performing methods, but surprisingly, a simple regression model outperforms almost half of the dedicated spatial deconvolution methods. Furthermore, we observe that the performance of all methods significantly decreased in datasets with highly abundant or rare cell types. Our results are reproducible in a Nextflow pipeline, which also allows users to generate synthetic data, run deconvolution methods and optionally benchmark them on their dataset (https://github.com/saeyslab/spotless-benchmark).

**\*For correspondence:**
yvan.saeys@ugent.be

**Competing interest:** The authors declare that no competing interests exist.

## eLife assessment

This study makes a **valuable** contribution to spatial transcriptomics by rigorously benchmarking cell-type deconvolution methods, assessing their performance across diverse datasets with a focus on biologically relevant, previously unconsidered aspects. The authors demonstrate the strengths of RCTD, cell2location, and SpatialDWLS for their performance, while also revealing the limitations of many methods when compared to simpler baselines. By implementing a full Nextflow pipeline, Docker containers, and a rigorous assessment of the simulator, this work offers robust insights that elevate the standards for future evaluations and provides a resource for those seeking to improve or develop new deconvolution methods. The thorough comparison and analysis of methods, coupled with a strong emphasis on reproducibility, provide **solid** support for the findings.

## Introduction

Unlike single-cell sequencing, spatial transcriptome profiling technologies can uncover the location of cells, adding another dimension to the data that is essential for studying systems biology, for example cell-cell interactions and tissue architecture (*Wagner et al., 2016*). These approaches can be categorized into imaging- or sequencing-based methods, each of them offering complementary advantages (*Asp et al., 2020*). As imaging-based methods use labeled hybridization probes to target specific genes, they offer subcellular resolution and high capture sensitivity, but are limited to a few hundreds

or thousands of genes (*Eng et al., 2019*; *Xia et al., 2019*). On the other hand, sequencing-based methods offer an unbiased and transcriptome-wide coverage by using capture oligonucleotides with a polydT sequence (*Rodriques et al., 2019*; *Ståhl et al., 2016*). These oligos are printed in clusters, or *spots*, each with a location-specific barcode that allows identification of the originating location of a transcript. While the size of these spots initially started at 100 μm in diameter, some recent sequencing technologies have managed to reduce their size to the subcellular level, thus closing the resolution gap with imaging technologies (*Chen et al., 2022*; *Cho et al., 2021*). However, these spots do not necessarily correspond to individual cells, and therefore, computational methods remain necessary to determine the cell-type composition of each spot.

Deconvolution and mapping are two types of cell type composition inference tools that can be used to disentangle populations from a mixed gene expression profile (*Longo et al., 2021*). In conjunction with the spatial dataset, a reference scRNA-seq dataset from an atlas or matched sequencing experiment is typically required to build cell-type-specific gene signatures. Deconvolution infers the proportions of cell types in a spot by utilizing a regression or probabilistic framework, and methods specifically designed for ST data often incorporate additional model parameters to account for spot-to-spot variability. On the other hand, mapping approaches score a spot for how strongly its expression profile corresponds to those of specific cell types. As such, deconvolution returns the proportion of cell types per spot, and mapping returns the probability of cell types belonging to a spot. Unless otherwise specified, we use the term 'deconvolution' to refer to both deconvolution and mapping algorithms in this study.

Although there have recently been multiple benchmarking studies (*Li et al., 2022*; *Yan and Sun, 2023*; *Li et al., 2023b*), several questions remain unanswered. First, the added value of algorithms specifically developed for the deconvolution of ST data has not been evaluated by comparing them to a baseline or bulk deconvolution method. Second, it is unclear which algorithms are better equipped to handle challenging scenarios, such as the presence of a highly abundant cell type throughout the tissue or the detection of a rare cell type in a single region of interest. Finally, the stability of these methods to variations in the reference dataset arising from changes in technology or protocols has not been assessed.

In this study, we address these gaps in knowledge and provide a comprehensive evaluation of 11 deconvolution methods in terms of performance, stability, and scalability (*Figure 1*). The tools include eight spatial deconvolution methods (cell2location *Kleshchevnikov and Shmatko, 2020*, DestVI *Lopez et al., 2022*, DSTG *Song and Su, 2021*, RCTD *Cable et al., 2022*, SpatialDWLS *Dong and Yuan, 2021*, SPOTlight *Elosua-Bayes et al., 2021*, stereoscope *Andersson et al., 2020*, and STRIDE *Sun et al., 2022*), one bulk deconvolution method (MuSiC *Wang et al., 2019*), and two mapping methods (Seurat *Stuart et al., 2019* and Tangram *Biancalani et al., 2021*). For all methods compared, we discussed with the original authors in order to ensure that their method was run in an appropriate setting and with good parameter values. We also compared method performance with two baselines: a 'null distribution' based on random proportions drawn from a Dirichlet distribution, and predictions from the non-negative least squares (NNLS) algorithm. We evaluated method performance on a total of 66 synthetic datasets (63 silver standards and 3 gold standards) and two application datasets. Our benchmarking pipeline is completely reproducible and accessible through Nextflow (github.com/ saeyslab/spotless-benchmark). Furthermore, each method is implemented inside a Docker container, which enables users to run the tools without requiring prior installation.

## Results

### *Synthspot* allows simulation of artificial tissue patterns

Synthetic spatial datasets are commonly generated by developers of deconvolution methods as part of benchmarking their algorithms against others. However, these synthetic datasets typically have spots with random compositions that do not reflect the reality of tissue regions with distinct compositions, such as layers in the brain. On the other hand, general-purpose simulators are more focused on other inference tasks, such as spatial clustering and cell-cell communication, and are usually unsuitable for deconvolution. For instance, generative models and kinetic models like those of scDesign3 (*Song and Wang, 2024*) and scMultiSim (*Li et al., 2023a*) are computationally intensive and unable to model entire transcriptomes. SRTSim (*Zhu, 2023*) focuses on modeling gene expression trends and does not

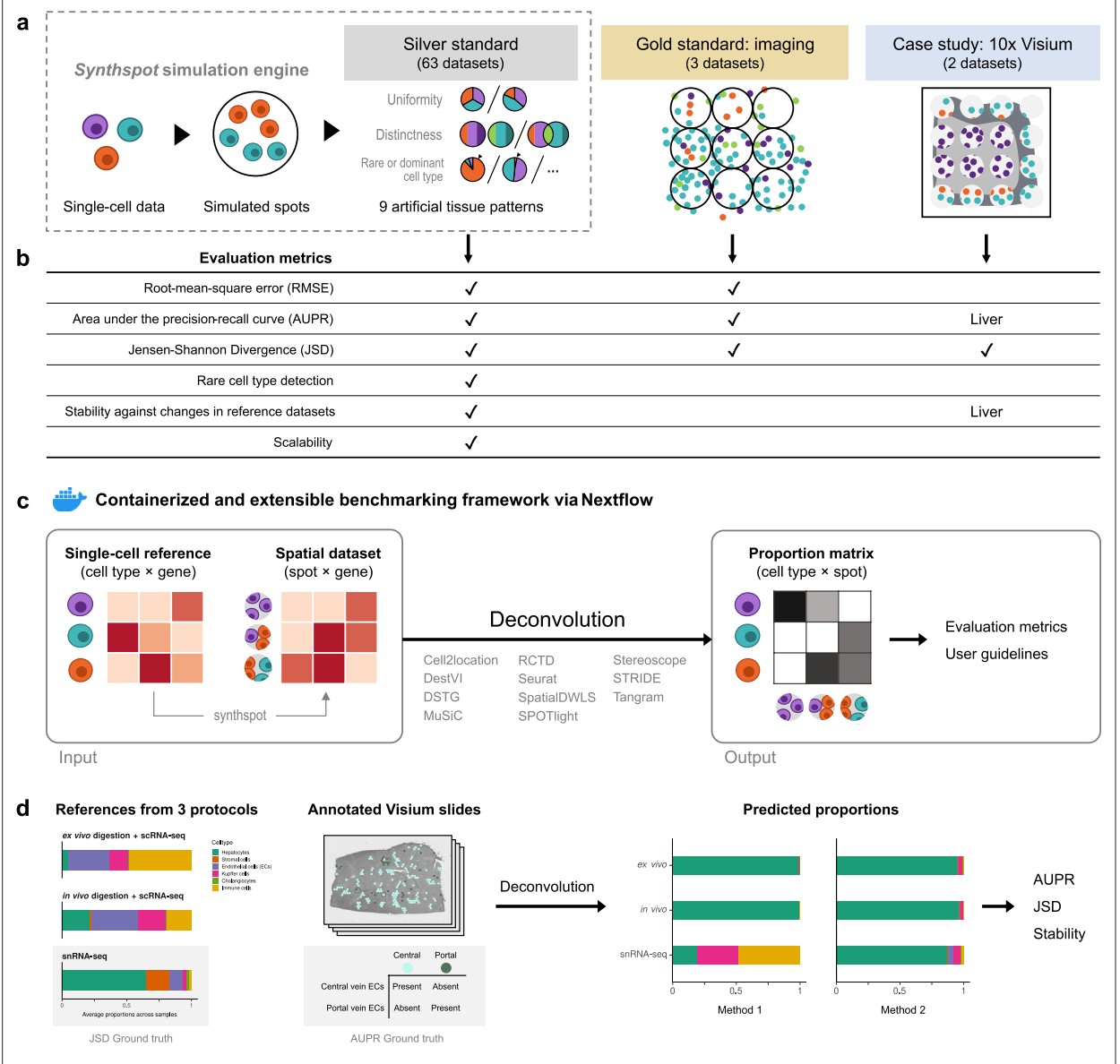

**Figure 1.** Overview of the benchmark. (**a**) The datasets used consist of silver standards generated from single-cell RNA-seq data, gold standards from imaging-based data, and two case studies on liver and melanoma. Our simulation engine *synthspot* enables the creation of artificial tissue patterns. (**b**) We evaluated deconvolution methods on three overall performance metrics (RMSE, AUPR, and JSD), and further checked specific aspects of performance, that is how well methods detect rare cell types and handle reference datasets from different sequencing technologies. For the case studies, the AUPR and stability are only evaluated on the liver dataset. (**c**) Our benchmarking pipeline is entirely accessible and reproducible through the use of Docker containers and Nextflow. (**d**) To evaluate performance on the liver case study, we leveraged prior knowledge of the localization and composition of cell types to calculate the AUPR and JSD. We also investigated method performance on three different sequencing protocols.

The online version of this article includes the following figure supplement(s) for figure 1:

**Figure supplement 1.** Overview of *synthspot* abundance patterns used in the study.

**Figure supplement 2.** Data generation scheme of silver standards.

**Figure supplement 3.** UMAP and violin plots of three of the seven scRNA-seq datasets used to generate silver standards.

**Figure supplement 4.** (Continuation of the previous figure supplement) UMAP and violin plots of four of the seven scRNA-seq datasets used to generate silver standards.

explicitly model tissue composition, while spaSim (*Feng et al., 2023*) only models tissue composition without gene expression. To overcome these limitations, we developed our own simulator called *synthspot* that can generate synthetic tissue patterns with distinct regions, allowing for more realistic simulations (https://github.com/saeyslab/synthspot; *Browaeys and Sang-aram, 2024*). We validated that our simulation procedure accurately models real data characteristics and that method performances are comparable between synthetic and real tissue patterns (Appendix 1).

Within a synthetic dataset, *synthspot* creates artificial regions in which all spots belonging to the same region have the same frequency priors. Frequency priors correspond to the likelihood in which cells from a cell type will be sampled, and therefore, spots within the same region will have similar compositions. These frequency priors are influenced by the chosen artificial tissue pattern, or *abundance pattern*, which determines the uniformity, distinctness, and rarity of cell types within and across regions (*Figure 1a*, *Figure 1—figure supplement 1*). For example, the *uniform* characteristic will sample the same number of cells for all cell types in a spot, while *diverse* samples differing number of cells. The *distinct* characteristic constrains a cell type to only be present in one region, while *overlap* allows it to be present in multiple regions. Additionally, the *dominant cell type* characteristic randomly assigns a dominant cell type that is at least 5–15 times more abundant than others in each spot, while *rare cell type* does the opposite to create a cell type that is 5–15 times less abundant. The different characteristics can be combined in up to nine different abundance patterns, each representing a plausible biological scenario.

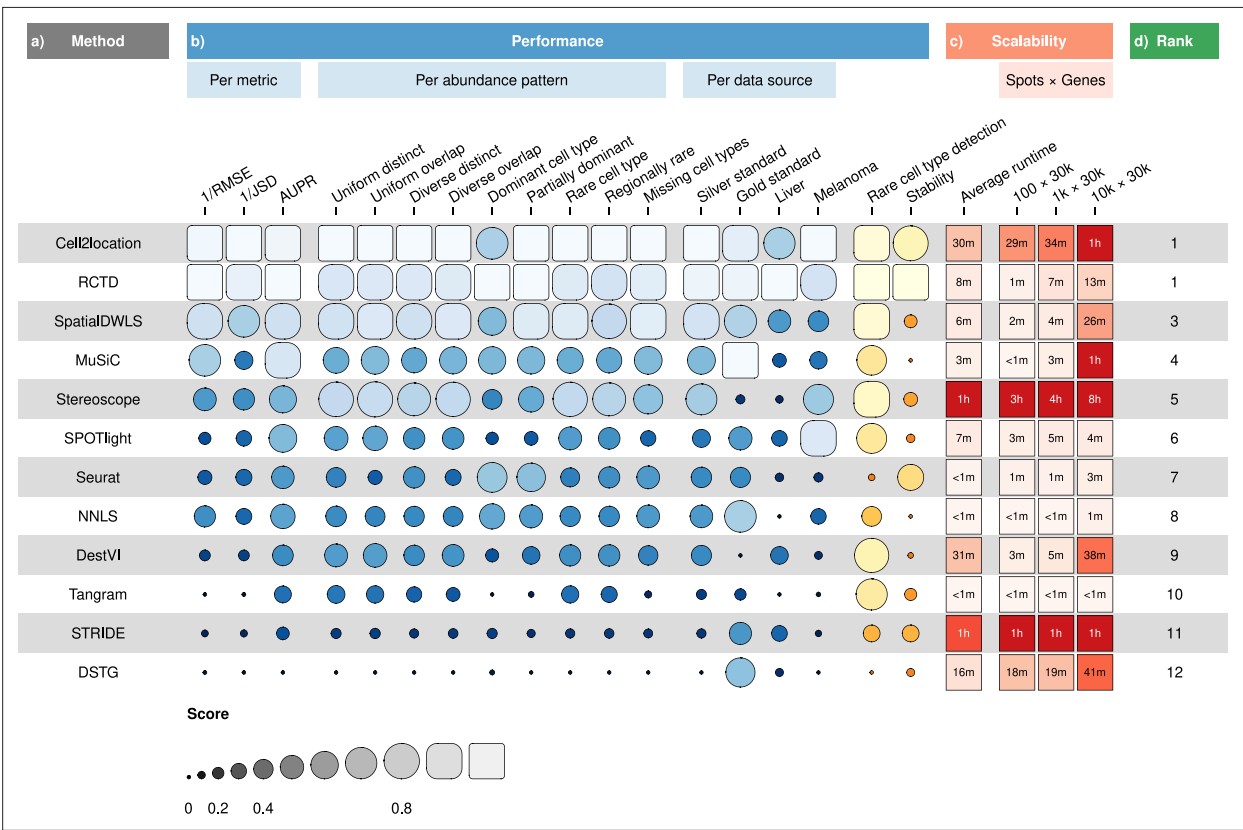

**Figure 2.** Overall results of the benchmark. (**a**) Methods ordered according to their overall rankings (**d**), determined by the aggregated rankings of performance and scalability. (**b**) Performance of each method across metrics, artificial abundance patterns in the silver standard, and data sources. The ability to detect rare cell types and stability against different reference datasets are also included. (**c**) Average runtime across silver standards and scalability on increasing dimensions of the spatial dataset.

The online version of this article includes the following source data for figure 2:

**Source data 1.** Raw data table of *Figure 2*.

## Cell2location and RCTD are the top performers in synthetic data

Our synthetic spatial datasets consist of 63 silver standards generated from *synthspot* and 3 gold standards generated from imaging data with single-cell resolution (*Supplementary file 1a-b*). We generated the silver standards using seven publicly available scRNA-seq datasets and nine abundance patterns. The scRNA-seq datasets consisted of four mouse brain tissues (cortex, hippocampus, and two cerebellum), mouse kidney, mouse skin cancer (melanoma), and human skin cancer (squamous cell carcinoma). Half of the cells from each scRNA-seq dataset were used to generate the synthetic spots and the other half used as the reference for deconvolution. This split was stratified by cell type. We generated 10 replicates for each of the 63 silver standards, with each replicate containing around 750 spots (*Figure 1—figure supplement 2*). For the gold standard, we used two seqFISH+ sections of mouse brain cortex and olfactory bulb (63 spots with 10,000 genes each) and one STARMap section of mouse primary visual cortex (108 spots with 996 genes; *Eng et al., 2019*; *Wang et al., 2018*). We summed up counts from cells within circles of 55 µm diameter, which are the size of spots in the 10x Visium commercial platform.

We evaluated method performance with the root-mean-square error (RMSE), area under the precision-recall curve (AUPR), and Jensen-Shannon divergence (JSD; Appendix 2). The RMSE measures how numerically accurate the predicted proportions are, the AUPR measures how well a method can detect whether a cell type is present or absent, and the JSD measure similarities between two probability distributions.

RCTD and cell2location were the top two performers across all metrics in the silver standards, followed by SpatialDWLS, stereoscope, and MuSiC (*Figure 2b*, *Figure 3a*). All other methods ranked worse than NNLS in at least one metric. For each silver standard, method rankings were determined using the median value across 10 replicates. We observed that method performances were more consistent between abundance patterns than between datasets (*Figure 3—figure supplements 1–3*). Most methods had worse performance in the two abundance patterns with a dominant cell type, and there was considerable performance variability between replicates due to different dominant cell types being selected in each replicate. Only RCTD and cell2location consistently outperformed NNLS in all metrics in these patterns (*Figure 3—figure supplements 4–5*).

For the gold standards, cell2location, MuSiC, and RCTD are the top three performers as well as the only methods to outrank NNLS in all three metrics (*Figure 3b*). As each seqFISH+ dataset consisted of seven field of views (FOVs), we used the average across FOVs as the representative value to be ranked for that dataset. Several FOVs were dominated by one cell type (*Figure 3—figure supplement 6*), which was similar to the dominant cell type abundance pattern in our silver standard. Consistent with the silver standard results, half of the methods did not perform well in these FOVs. In particular, SPOTlight, DestVI, stereoscope, and Tangram tended to predict less variation between cell type abundances. DSTG, SpatialDWLS, and Seurat predicted the dominant cell type in some FOVs but did not predict the remaining cell type compositions accurately. Most methods performed worse in the STARMap dataset except for SpatialDWLS, SPOTlight, and Tangram.

## Detecting rare cell types remains challenging even for top-performing methods

Lowly abundant cell types often play an important role in development or disease progression, as in the case of stem cells and progenitor cells or circulating tumor cells (*Jindal et al., 2018*). As the occurrence of these cell types are often used to create prognostic models of patient outcomes, the accurate detection of rare cell types is a key aspect of deconvolution tools (*Ali et al., 2016*; *Sato et al., 2005*). Here, we focus on the two *rare cell type* patterns in our silver standard (*rare cell type diverse* and *regional rare cell type diverse*), in which a rare cell type is 5–15 times less abundant than other cell types in all or one synthetic region, respectively (*Figure 1—figure supplement 1*). This resulted in 14 synthetic datasets (seven scRNA-seq datasets × two abundance patterns) with 10 replicates each. We only evaluated methods based on the AUPR of the rare cell type, using the average AUPR across the 10 replicates as the representative value for each dataset. We did not include the RMSE and JSD or consider other cell types, because in prognostic models, the presence of rare cell types is often of more importance than the magnitude of the abundance itself. Therefore, it is more relevant that methods are able to correctly rank spots with and without the rare cell type.

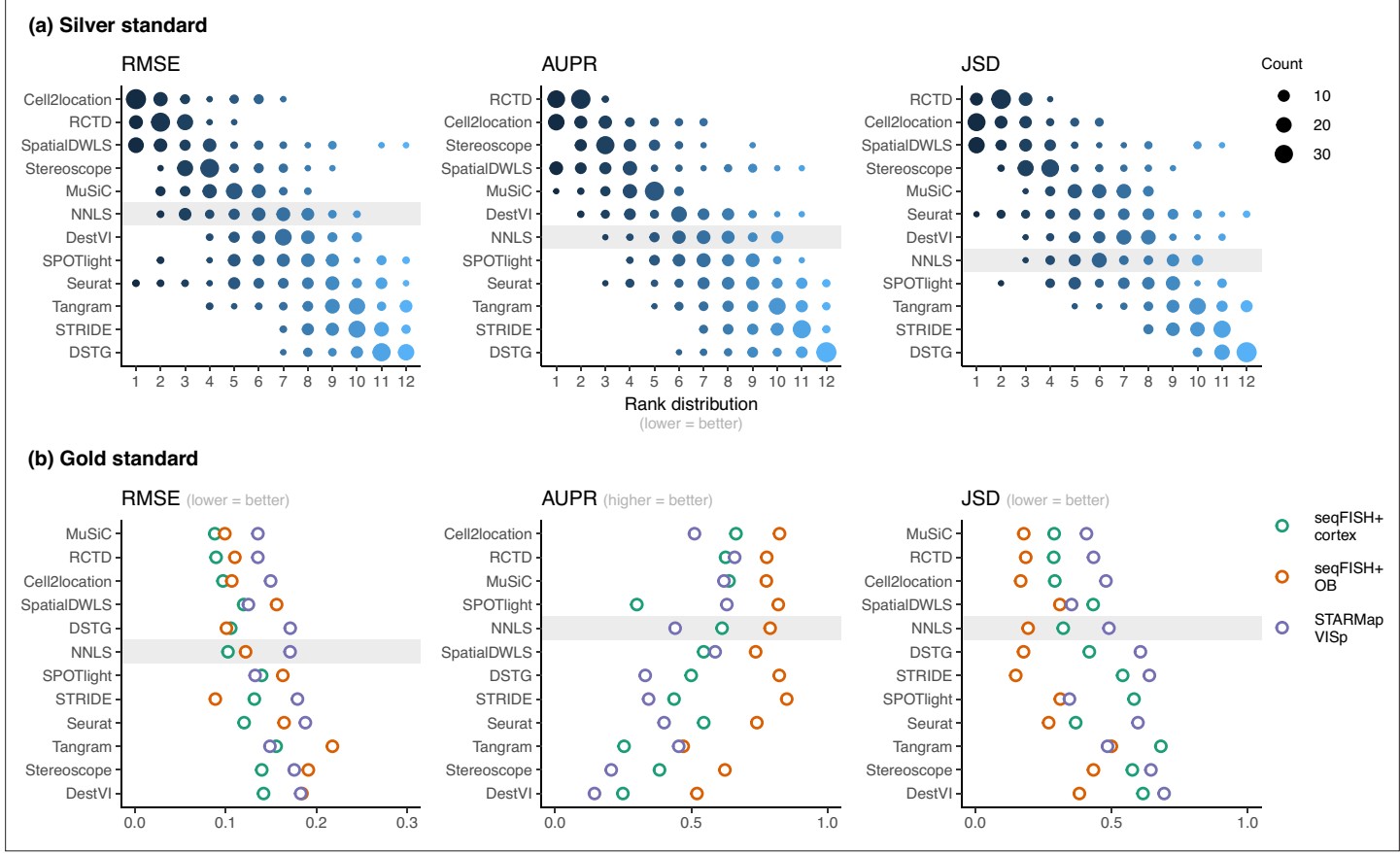

**Figure 3.** Method performance on synthetic datasets, evaluated using root-mean-squared error (RMSE), area under the precision-recall curve (AUPR), and Jensen-Shannon divergence (JSD). Non-negative least squares (NNLS) is shaded as a baseline algorithm. Methods are ordered based on the summed ranks across all 63 and three datasets, respectively. (**a**) The rank distribution of each method across all 63 silver standards, based on the best median value across ten replicates for that standard. (**b**) Gold standards of two seqFISH+ datasets and one STARMap dataset. We took the average over seven field of views for the seqFISH+ dataset.

The online version of this article includes the following source data and figure supplement(s) for figure 3:

**Source data 1.** Raw data table of *Figure 3*.

**Figure supplement 1.** Boxplots of root-mean-squared error (RMSE) across ten replicates for each silver standard dataset (row) and abundance pattern (column).

**Figure supplement 2.** Boxplots of the area under the precision-recall curve (AUPR) across ten replicates for each silver standard dataset (row) and abundance pattern (column).

**Figure supplement 3.** Boxplots of Jensen-Shannon divergence across ten replicates for each silver standard dataset (row) and abundance pattern (column).

**Figure supplement 4.** Summed rank plots across all silver standard datasets for each abundance pattern (column) and metric (row).

**Figure supplement 5.** (Continuation of the previous figure supplement) Summed rank plots across all silver standard datasets for each abundance pattern (column) and metric (row).

**Figure supplement 6.** Summed abundances across all spots for each gold standard dataset.

In line with our previous analysis, RCTD and cell2location were also the best at predicting the presence of lowly abundant cell types (*Figure 4a*). There is a strong correlation between cell type abundance and AUPR, as clearly demonstrated when plotting the precision-recall curves of cell types with varying proportions (*Figure 4b*). While most methods can detect cell types with moderate or high abundance, they usually have lower sensitivity for rare cell types, and hence lower AUPRs. Upon visual inspection of precision-recall curves at decreasing abundance levels, we found a similar pattern across all silver standards (*Figure 4—figure supplement 1*). Nonetheless, we also observe that the AUPR is

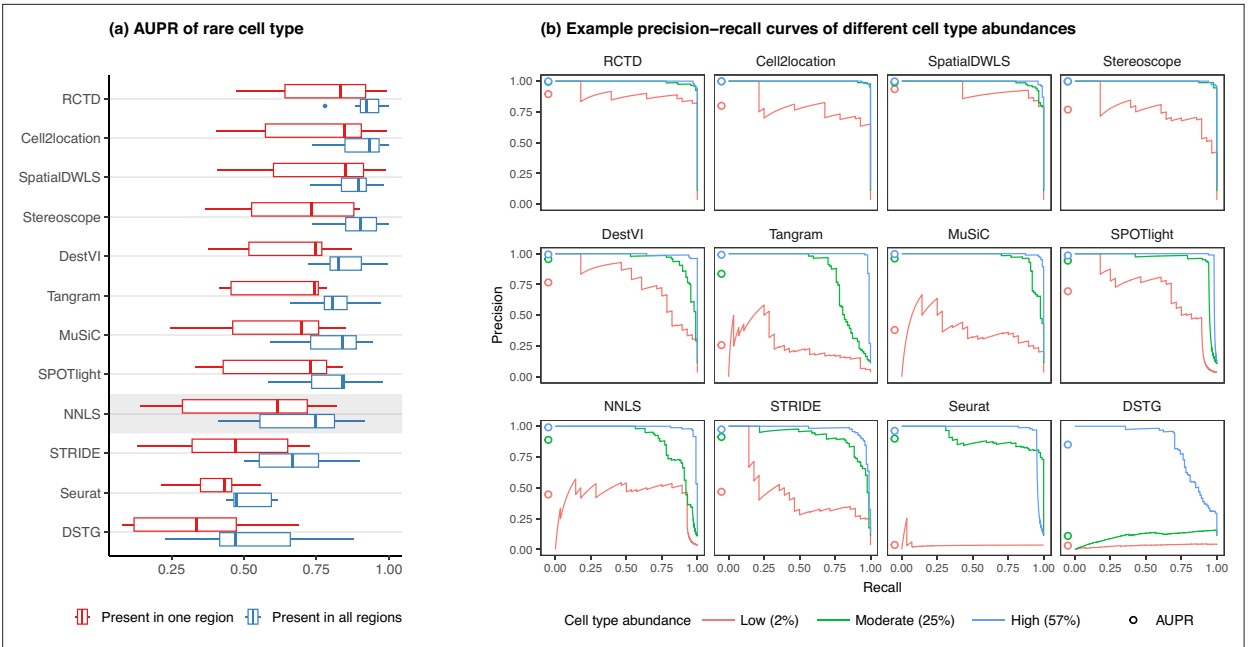

**Figure 4.** Detection of the rare cell type in the two *rare cell type* abundance patterns. (**a**) Area under the precision-recall curve (AUPR) across the seven scRNA-seq datasets, averaged over 10 replicates. Methods generally have better AUPR if the rare cell type is present in all regions compared to just one region. (**b**) An example on one silver standard replicate demonstrates that most methods can detect moderately and highly abundant cells, but their performance drops for lowly abundant cells.

The online version of this article includes the following source data and figure supplement(s) for figure 4:

**Source data 1.** Raw data table of *Figure 4*.

**Figure supplement 1.** Evaluating the AUPR as a function of cell type abundance.

substantially lower if the rare cell type is only present in one region and not across the entire tissue, indicating that prevalence is also an important factor in detection.

## Technical variability between reference scRNA-seq and spatial datasets can significantly impact predictions

Since deconvolution predictions exclusively rely on signatures learned from the scRNA-seq reference, it should come as no surprise that the choice of the reference dataset has been shown to have the largest impact on bulk deconvolution predictions (*Vallania et al., 2018*). Hence, deconvolution methods should ideally also account for platform effects, that is the capture biases that may occur from differing protocols and technologies being used to generate scRNA-seq and ST data.

To evaluate the stability of each method against reference datasets from different technological platforms, we devised the *inter*-dataset scenario, where we provided an alternative reference dataset to be used for deconvolution, in contrast to the *intra*-dataset analysis done previously, where the same reference dataset was used for both spot generation and deconvolution. We tested this on the brain cortex (SMART-seq) and two cerebellum (Drop-seq and 10x Chromium) silver standards. For the brain cortex, we used an additional 10x Chromium reference from the same atlas (*Yao et al., 2021*). To measure stability, we computed the JSD between the proportions predicted from the intra- and inter-dataset scenario.

Except for MuSiC, we see that methods with better performance–cell2location, RCTD, SpatialDWLS, and stereoscope–were also more stable against changing reference datasets (*Figure 5*). Out of these four, only SpatialDWLS did not account for a platform effect in its model definition. Cell2location had the most stable predictions, ranking first in all three datasets, while both NNLS and MuSiC were consistently in the bottom three. For the rest of the methods, stability varied between datasets. DestVI performed well in the cerebellum datasets but not the brain cortex, and SPOTlight had the opposite

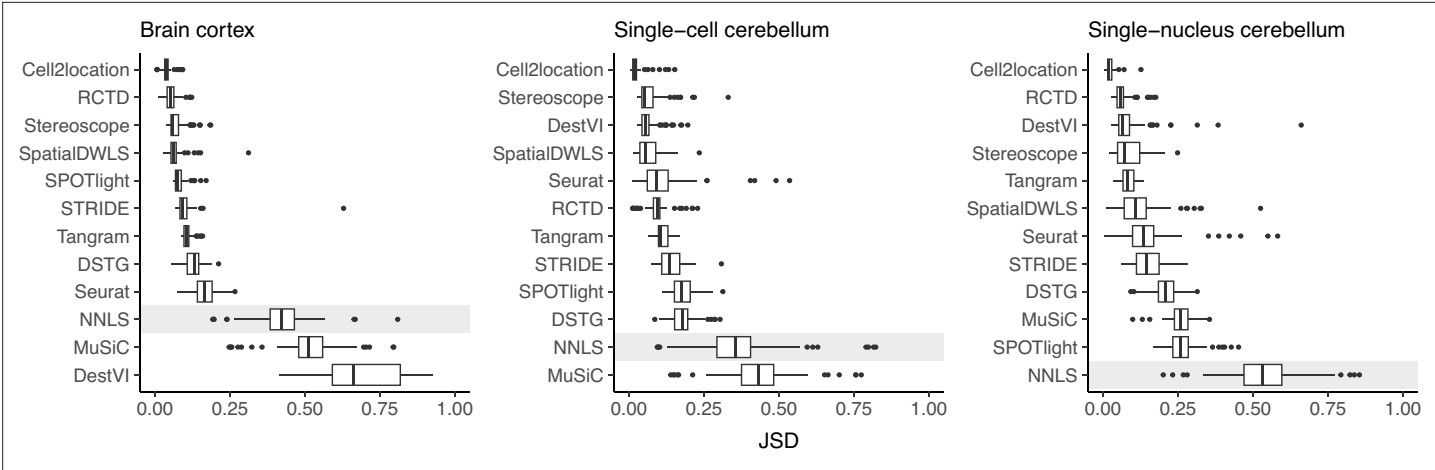

**Figure 5.** Prediction stability when using different reference datasets. For each synthetic dataset (total n=90, from nine abundance patterns with ten replicates each), we computed the Jensen-Shannon divergence between cell type proportions obtained from two different reference datasets.

The online version of this article includes the following source data and figure supplement(s) for figure 5:

**Source data 1.** Raw data table of *Figure 5*.

**Figure supplement 1.** Changes in performance metrics when using a different reference dataset, from a matched to an unmatched reference (i.e., intra-dataset vs inter-dataset scenario).

pattern. As expected, deconvolution performance also generally worsens in the inter-dataset scenario (*Figure 5—figure supplement 1*).

## Evaluation of methods on public Visium datasets validate results on synthetic data

While synthetic datasets provide valuable benchmarks, it is crucial to validate deconvolution methods using real sequencing-based ST data, as they may exhibit distinct characteristics. To achieve this, we leveraged 10x Visium datasets from two mouse tissues, namely liver and skin cancer (melanoma). These datasets were chosen due to the availability of ground truth knowledge regarding cell proportions and localization, as elaborated further below.

### Liver

The liver is a particularly interesting case study due to its tissue pattern, where hepatocytes are highly abundant and constitute more than 60% of the tissue. This characteristic allows us to compare method performance with those of the *dominant cell type* abundance pattern from the silver standard, which was challenging for most methods. Here, we use the four Visium slides and single-cell dataset from the liver cell atlas of *Guilliams et al., 2022*; *Supplementary file 1c*. The single-cell dataset was generated from three different experimental protocols–scRNA-seq following ex vivo digestion, scRNA-seq following in vivo liver perfusion, and single-nucleus RNA-seq (snRNA-seq) on frozen liver—additionally enabling us to assess method stability on different reference datasets.

We assessed method performance using AUPR and JSD by leveraging prior knowledge of the localization and composition of cell types in the liver. Although the true composition of each spot is not known, we can assume the presence of certain cell types in certain locations in the tissue due to the zonated nature of the liver. Specifically, we calculated the AUPR of portal vein and central vein endothelial cells (ECs) under the assumption that they are present only in their respective venous locations. Next, we calculated the JSD of the expected and predicted cell type proportions in a liver sample. The expected cell type proportions were based on snRNA-seq data (*Figure 6—figure supplement 1*), which has been shown to best recapitulate actual cell compositions observed by confocal microscopy (*Guilliams et al., 2022*). We ran each method on a combined reference containing all three experimental protocols (ex vivo scRNA-seq, in vivo scRNA-seq, and snRNA-seq), as well as on each protocol separately. To ensure consistency, we filtered each reference dataset to only include the nine cell types that were common to all three protocols.

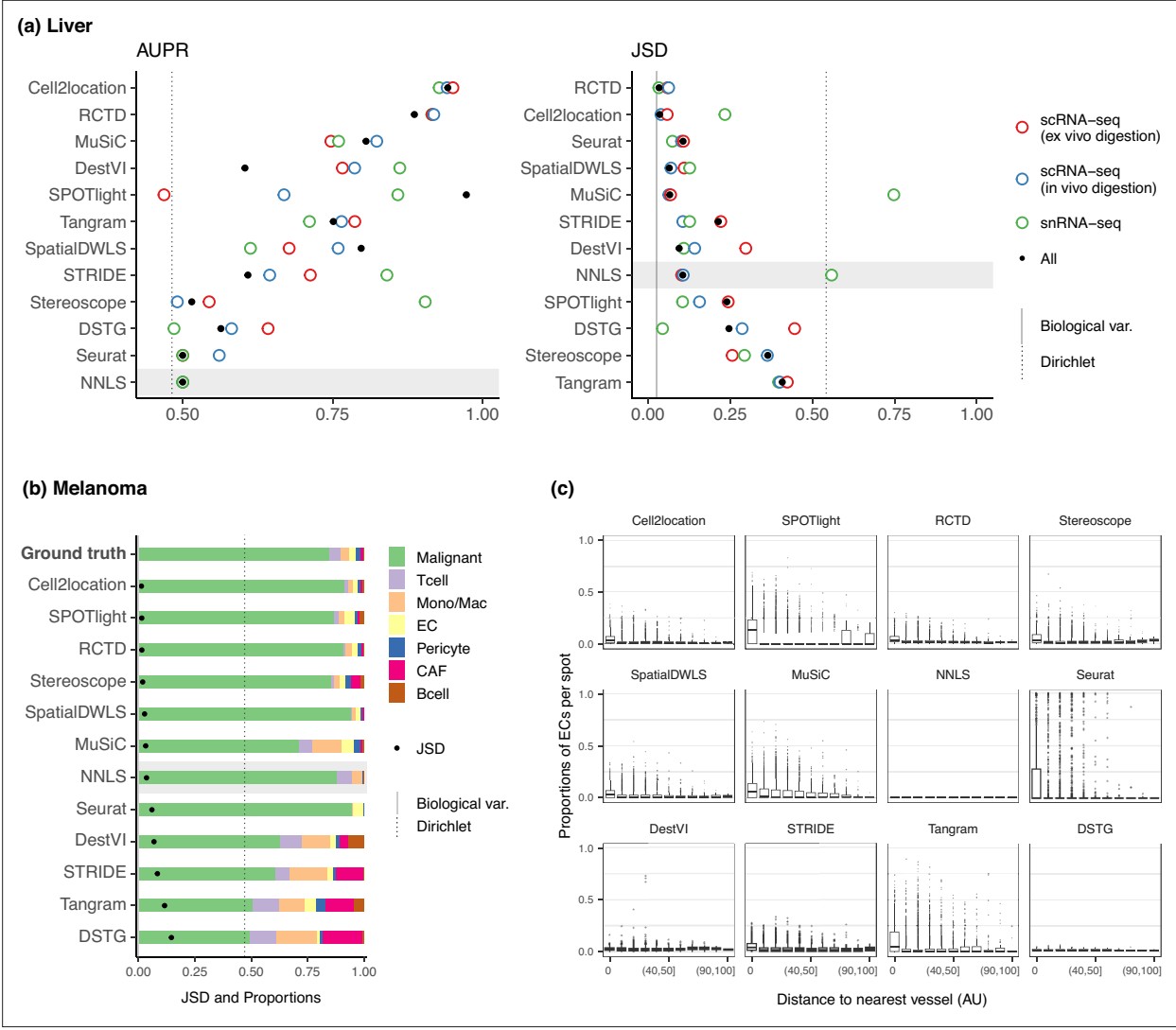

**Figure 6.** Method performance on two Visium case studies. (**a**) In the liver case study, the AUPR was calculated using the presence of portal/central vein endothelial cells in portal and central veins, and the JSD was calculated by comparing predicted cell type proportions with those from snRNA-seq. All reference datasets contain nine cell types. Biological variation refers to the average pairwise JSD between four snRNA-seq samples. Methods are ordered based on the summed rank of all data points. (**b**) For melanoma, the JSD was calculated between the predicted cell type proportions and those from Molecular Cartography (bold). Biological variation refers to the JSD between the two Molecular Cartography samples. (**c**) Relationship between the proportions of endothelial cells predicted per spot and their distance to the nearest blood vessel (in arbitrary units, AU), where zero denotes a spot annotated as a vessel. An inverse correlation can be observed more clearly in better-performing methods.

The online version of this article includes the following source data and figure supplement(s) for figure 6:

**Source data 1.** Raw data table of *Figure 6*.

**Figure supplement 1.** Comparison of cell type compositions between three sequencing protocols in the mouse liver atlas from *Guilliams et al., 2022*.

**Figure supplement 2.** Predicted cell type abundances averaged across all four Visium slides from the liver atlas.

**Figure supplement 3.** Concordance of method performance between the synthetic dataset (generated using synthspot's *dominant cell type* abundance pattern) and the Visium dataset.

**Figure supplement 4.** The predicted abundance of central vein and portal vein endothelial cells (ECs) for each spot in one liver Visium slide.

**Figure supplement 5.** Stability of predicted proportions when using three different protocols from the liver atlas as reference for deconvolution.

**Figure supplement 6.** The ground truth and predicted cell type proportions of the melanoma dataset.

RCTD and cell2location were the top performers for both AUPR and JSD (*Figure 6a*), achieving a JSD comparable to those of biological variation, that is the average pairwise JSD between four snRNA-seq samples. In contrast, Tangram, DSTG, SPOTlight, and stereoscope had higher JSD values than those of NNLS. Except for SPOTlight, these three methods were not able to accurately infer the overabundance of the dominant cell type, with stereoscope and Tangram predicting a uniform distribution of cell types (*Figure 6—figure supplement 2*). This corresponds to what we observed in the *dominant cell type* pattern of the silver standard and certain FOVs in the gold standard. To further substantiate this, we used the ex vivo scRNA-seq dataset to generate 10 synthetic datasets with the *dominant cell type* abundance pattern. Then, we used the remaining two experimental protocols separately as the reference for deconvolution. Indeed, the method rankings followed the overall pattern, and a comparison of the JSD values between synthetic and real Visium data revealed a strong correlation (*Figure 6—figure supplement 3*).

Nonetheless, there were some inconsistencies between the AUPR and JSD rankings, specifically with Seurat and SpatialDWLS ranking highly for JSD and lowly for AUPR. This is because the JSD is heavily influenced by the dominant cell type in the dataset, such that even when predicting only the dominant cell type per spot, Seurat still performed well in terms of JSD. However, it was unable to predict the presence of portal and central vein ECs in their respective regions (*Figure 6—figure supplement 4*). Therefore, complementary metrics like AUPR and JSD must both be considered when evaluating methods.

We observed that using the combination of the three experimental protocols as the reference dataset did not necessarily result in the best performance, and it was often possible to achieve better or similar results by using a reference dataset from a single experimental protocol. The best reference varied between methods, and most methods did not exhibit consistent performance across all references. Interestingly, Cell2location, MuSiC, and NNLS had much higher JSD when using the snRNA-seq data as the reference, while RCTD and Seurat had the lowest JSD on the same reference. To further evaluate the stability of the methods, we calculated the JSD between proportions predicted with different reference datasets. RCTD and Seurat showed the lowest JSD, indicating higher stability (*Figure 6—figure supplement 5*). Finally, we examined the predicted proportions when using the entire atlas without filtering cell types, which contains all three protocols and 23 cell types instead of only the nine common cell types (*Figure 6—figure supplement 2*). The additional 14 cell types made up around 20% of the ground truth proportions. While RCTD, Seurat, SpatialDWLS, and MuSiC retained the relative proportions of the nine common cell types, the rest predicted substantially different cell compositions.

## Melanoma

Melanoma poses a significant challenge in both therapeutic and research efforts due to its high degree of heterogeneity and plasticity. In a recent study, *Karras et al., 2022* investigated the cell state diversity of melanoma by generating single-cell and ST data from a mouse melanoma model (*Supplementary file 1d*). Among others, the spatial data consists of three individual tumor sections profiled by 10x Visium, as well as 33 regions of interest from two tumor samples profiled by Molecular Cartography (MC), an imaging-based technology that can profile up to 100 genes at subcellular resolution. Using a custom processing pipeline, we obtained cell type annotations—and subsequently the cell type proportions—for each of the MC samples. These cell type proportions were consistent across different sections and samples, and were used as the ground truth for deconvolution (*Figure 6—figure supplement 6*). We aggregated the predicted proportions of the seven malignant cell states in the Visium slides, as we could not reliably annotate these cell states in the MC dataset.

To assess method performance, we calculated the average pairwise JSDs between the two MC samples and three Visium slides. Cell2location, SPOTlight, and RCTD were the best performers with JSDs of around 0.01 (*Figure 6b*). With the exception of SPOTlight performing well in this dataset, the rankings of the remaining methods followed the usual trend in the silver standards, with NNLS outperforming Seurat, DestVI, STRIDE, and DSTG. Additionally, we sought to corroborate these findings through a more qualitative approach by leveraging the blood vessel annotations provided in the original study. Given that by definition, endothelial cells form the linings of blood vessels, we visualized the relationship between EC abundance and the distance to the nearest vessel (*Figure 6c*). Although the majority of spots were predicted to have no ECs, a fraction of spots exhibited the expected

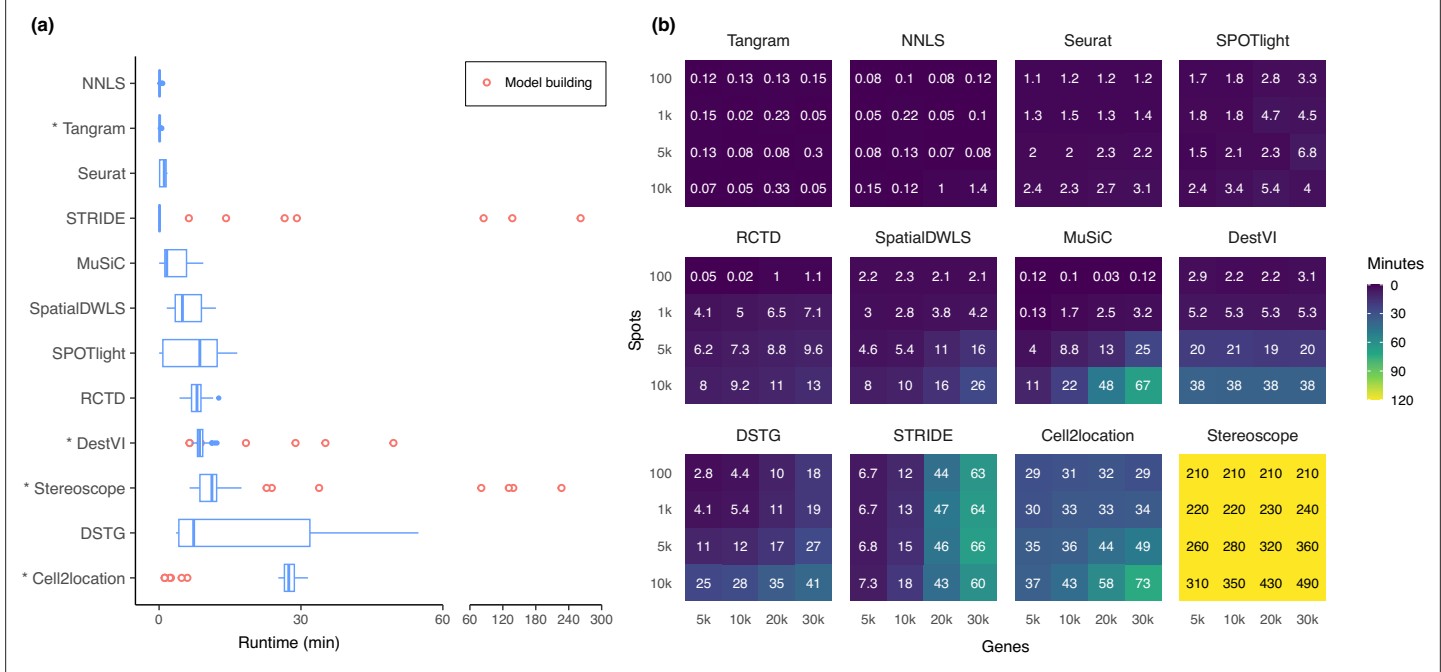

**Figure 7.** Runtime and scalability of each method. (**a**) Runtime over the 63 silver standards (three replicates each). Methods are ordered by total runtime. Asterisks indicate when GPU acceleration has been used. Cell2location, stereoscope, DestVI, and STRIDE first build a model for each single-cell reference (red points), which can be reused for all synthetic datasets derived from that reference. (**b**) Method scalability on increasing dimensions of the spatial dataset. For model-based methods, the model building and fitting time were summed. Methods are ordered based on total runtime.

The online version of this article includes the following source data for figure 7:

**Source data 1.** Raw data table of *Figure 7*.

trend where ECs were more abundant the closer the spot was to a blood vessel. This trend was more discernible for higher ranked methods, while the lower-ranked ones either showed no correlation (NNLS, DestVI, DSTG) or had noisy predictions (Seurat, Tangram).

## Runtime and scalability

Most methods are able to deconvolve the silver standard datasets in less than 30 min and only had slight variability in the runtime (*Figure 7a*). Model-based methods—DestVI, stereoscope, cell2location, and STRIDE—have the advantage that a model built from a single-cell reference can be reused for all synthetic datasets derived from that reference (i.e. the nine abundance patterns ×10 replicates). This typically reduces the runtime needed to fit the model on the synthetic datasets. However, these methods tend to be more computationally intensive and are strongly recommended to be run with GPU acceleration. This was not implemented in STRIDE and could explain its longer runtime during model building.

Next, we investigated method scalability by varying the dimensions of the spatial dataset (*Figure 7b*). We tested 16 combinations of spots (100, 1 k, 5 k, 10 k) and genes (5 k, 10 k, 20 k, 30 k), while the single-cell dataset was kept at 5 k cells and 30 k genes. Here, we considered both the model building and fitting step in the runtime (when applicable). Tangram, Seurat, and SPOTlight had a constant runtime across all dimensions, each deconvolving the largest dataset ($3 \times 10^8$ elements in total) in less than ten minutes. Other methods had increased runtime both when spots and genes were increased, except for DestVI which was only affected by the number of spots, and STRIDE by the number of genes. This was because by default, DestVI only uses 2000 highly variable genes for deconvolution. Similarly, the scalability of RCTD, SpatialDWLS, Tangram, and SPOTlight are due to the fact that they only make use of differentially expressed or cell type-specific marker genes. Stereoscope was the least scalable by far, taking over 8 hr for the largest dataset, and 3.5 hr for the smallest one. Note that many methods allow for parameters that can greatly affect runtime, such as the number of training epochs and/or the number of genes to use (*Supplementary file 2*). For instance, here we

have used all genes to run stereoscope (default parameters), but the authors have suggested that it is possible to use only 5000 genes in the model and maintain the same performance.

## Discussion

In this study, we performed a thorough investigation of various biologically relevant and previously unconsidered aspects related to deconvolution. We evaluated the performance of 11 deconvolution methods on 63 silver standards, 3 gold standards, and 2 Visium datasets using three complementary metrics. We also incorporated two baselines for every analysis, the NNLS algorithm and null distribution proportions from a Dirichlet distribution. In the case studies, we demonstrated two approaches for evaluating deconvolution methods in datasets lacking an absolute ground truth. These approaches include using proportions derived from another sequencing or spatial technology as a proxy, and leveraging spot annotations, for example zonation or blood vessel annotations, that typically have already been generated for a separate analysis.

Our findings indicate that RCTD, cell2location, and SpatialDWLS were the highest ranked methods in terms of performance, consistent with previous studies (*Li et al., 2022*; *Yan and Sun, 2023*). However, we also found that over half of the methods did not outperform the baseline (NNLS) and bulk deconvolution method (MuSiC). These results were consistent across the silver and gold standards, as well as the liver and melanoma case studies, demonstrating the generalizability and applicability of our simulation and benchmarking framework. We also found that the abundance pattern of the tissue and the reference dataset used had the most significant impact on method performance. Even top-performing methods struggled with challenging abundance patterns, such as when a rare cell type was present in only one region, or when a highly dominant cell type masks the signature of less abundant ones. Furthermore, different reference datasets could result in substantially different predicted proportions. Methods that accounted for technical variability in their models, such as cell2location and RCTD, were more stable to changes in the reference dataset than those that did not, such as SpatialDWLS.

Regarding the reference dataset, the number of genes per cell type (which is generally correlated to the sequencing depth) seems to have a significant impact on the performance of deconvolution methods. We observed that methods were less accurate in datasets with fewer genes per cell type (*Figure 1—figure supplements 3–4*). For example, all methods performed better in the snRNA-seq cerebellum dataset, which had the same number of cell types as the scRNA-seq cerebellum dataset, but on average 1000 more genes per cell type. The kidney dataset was the most challenging for all methods, with most of its 16 cell types having less than 1000 genes. This was evident from the RMSE and JSD scores that were relatively closer to the null distribution than in other datasets (*Figure 3—figure supplements 1 and 3*). In contrast, the 18 cell types in the brain cortex dataset had an average of 3000–9000 features, leading to much better performance for most methods compared to the kidney dataset despite having more cell types. This trend was also observed in the STARMap gold standard dataset, which consisted of only 996 genes. Most methods performed worse in the STARMap dataset except for SpatialDWLS, SPOTlight, and Tangram. Since these three methods only use marker genes for deconvolution, this may explain why the small number of genes (most of which were already marker genes) did not affect them as much.

In addition to performance, the runtime and scalability of a method is also a factor to consider. Although most runtimes were comparable on our silver standards, we have made use of GPU acceleration for Tangram, DestVI, stereoscope, and cell2location. As this might not be available for some users, using these methods on the CPU might require training on fewer epochs or selecting only a subset of genes. With all these factors in consideration, we recommend RCTD as a good starting point. In addition to being one of the best and fastest methods, it also allows CPU parallelization (i.e. using multiple cores) for users that may not have access to a GPU.

As a general guideline, we recommend comparing the result of multiple deconvolution methods, especially between cell2location and RCTD. If the predictions are highly contradictory, the reference dataset may not be of sufficiently good quality. We recommend obtaining scRNA-seq and spatial data from the same sample to reduce biological variability, as there will always be technical variability across platforms. This can also ensure that the same cell types will be present in both datasets (*Longo et al., 2021*). If that is not possible, use a reference dataset that has sufficient sequencing depth (at least more than 1000 genes per cell type), preferably from a single platform. In addition to checking

the sequencing depth, our simulator *synthspot* can also be used to evaluate the quality of the reference dataset. As we have demonstrated in the liver case study, users can generate synthetic spatial datasets with an abundance pattern that best resembles their tissue of interest. With a high-quality reference, both cell2location and RCTD should be able to achieve an AUPR close to one and JSD close to zero. Our Nextflow pipeline seamlessly integrates the complete workflow of synthetic data generation, deconvolution, and performance evaluation.

As spatial omics is still an emerging field, the development of new deconvolution methods can be anticipated in the future. Our benchmark provides a reliable and reproducible evaluation framework for current and upcoming deconvolution tools, making it a valuable resource for researchers in the field.

## Methods
### Synthspot
We generated synthetic spatial datasets using the *synthspot* R package, whose spot generation process is modeled after that of SPOTlight (*Elosua-Bayes et al., 2021*). This simulator generates synthetic spot data by considering the gene expression of one spot as a mixture of the expression of *n* cells, with *n* being a random number between 2 and 10 that is sampled from a uniform distribution. To generate one spot, the simulator samples *n* cells from the input scRNA-seq data, sums their counts gene by gene, and downsamples the counts. The target number of counts for downsampling is picked from a normal distribution with mean and standard deviation of 20,000±5000 by default, but these values can be changed by the user. To mimic biological tissue, *synthspot* generates artificial regions, or groups of spots with similar cell type compositions (*Figure 1—figure supplement 1*). The prior distribution of each cell type in each region is influenced by the selected *abundance pattern*, called *dataset type* in the package (Appendix 1).

### Method execution
An overview of the 11 methods can be found in *Supplementary file 2*. As input, all methods require a reference scRNA-seq dataset with cell type annotations along with a spatial dataset. We first ran the methods based on the provided tutorials, and later reached out to the original authors of each method for additional recommendations. For most methods, we explored different parameters and selected ones that resulted in the best performance. The specifics on how each method was run and the final parameters can be found in **Appendix 3**. Unless stated otherwise, these parameters were applied on all datasets used in this study.

We implemented a Nextflow pipeline (*Di Tommaso et al., 2017*) to run the methods concurrently and compute their performance. For reproducibility, each method was installed inside a Docker container and can be run either using Docker or Apptainer (formerly known as Singularity). Our pipeline can be run in three modes: (1) *run_standard* to reproduce our benchmark, (2) *run_dataset* to run methods on a given scRNA-seq and spatial dataset, and (3) *generate_and_run* to also generate synthetic spatial data from the given scRNA-seq dataset. Our pipeline can accept both Seurat (.rds) and AnnData (.h5ad) objects as input, and the proportions are output as a tab-separated file. Parameter tuning has also been implemented.

All methods were deployed on a high-performance computing cluster with an Intel Xeon Gold 6140 processor operating at 2.3 GHz, running a RHEL8 operating system. We made use of GPU acceleration whenever possible using a NVIDIA Volta V100 GPU. We ran all methods with one core and 8 GB memory, dynamically increasing the memory by 8 GB if the process failed.

### Datasets
Gold
The seqFISH+ dataset consists of two mouse brain tissue slices (cortex and olfactory bulb) with seven field of views (FOVs) per slice (*Eng et al., 2019*). Ten thousand genes were profiled for each FOV. Each FOV had a dimension of 206 µm × 206 µm and a resolution of 103 nm per pixel. We simulated Visium spots of 55 µm diameter and disregarded the spot-to-spot distance, resulting in nine spots per FOV and 126 spots for the entire dataset. The FOVs had varying number of cells and cell types per simulated spot (*Supplementary file 1a*). We created a reference dataset for each tissue slice using the

combined single-cell expression profiles from all seven FOVs. There were 17 cell types in the cortex tissue section and nine cell types in the olfactory bulb tissue section.

The STARMap dataset of mouse primary visual cortex (VISp) is 1.4 mm × 0.3 mm and contains 1020 genes and 973 cells (*Wang et al., 2018*). We generated Visium-like spots as above, assuming that 1 pixel = 810 nm, as this information was not provided in the original paper. We removed any spot with hippocampal cell types, and removed unannotated and Reln cells from spots, resulting in 108 spots comprising 569 cells and 12 cell types in total. We used the VISp scRNA-seq dataset from the Allen Brain Atlas as the reference (*Tasic et al., 2016*). After filtering for common genes, 996 genes remained.

## Silver

We used six scRNA-seq datasets and one snRNA-seq dataset for the generation of silver standards (*Supplementary file 1b*). All datasets are publicly available and already contain cell type annotations. We downsized each dataset by filtering out cells with an ambiguous cell type label and cell types with less than 25 cells, and kept only highly variable genes (HVGs) that are expressed in at least 10% of cells from one cell type. We set this threshold at 25% for the brain cortex dataset. We further downsampled the kidney dataset to 15,000 cells, and the melanoma dataset to 20,000 cells. For the two cerebellum datasets, we integrated them following the method of *Stuart et al., 2019*, performed the same gene filtering approach on the integrated dataset, and only retained common cell types.

We generated the synthetic datasets using nine out of the 17 possible abundance patterns in *synthspot*. For each scRNA-seq dataset, we generated 10 replicates of each abundance pattern. For each replicate, we ran *generate_synthetic_visium* with five regions (*n_regions*), with minimally 100 spots and maximally 200 spots per region (*n_spots_min, n_spots_max*), and target counts of 20,000±5,000 per spot (*visium_mean,* and *visium_sd*). There were on average 750 spots per replicate.

## Liver

We downloaded single-cell and spatial datasets of healthy mice from the liver cell atlas (*Guilliams et al., 2022*; *Supplementary file 1c*). The single-cell data contains 185,894 cells and 31,053 genes from three experimental protocols: scRNA-seq following ex vivo digestion, scRNA-seq following in vivo liver perfusion, and snRNA-seq on frozen liver. We used the finer annotation of CD45$^-$ cells, which among others, subclustered endothelial cells into portal vein, central vein, lymphatic, and liver sinusoidal endothelial cells. We only retained cell types where at least 50 cells are present in each protocol, resulting in nine common cell types. The spatial data consisted of four Visium slides, containing on average 1440 spots. Each spot has been annotated by the original authors as belonging to either the central, mid, periportal, or portal zone. This zonation trajectory was calculated based on hepatocyte zonation markers.

We deconvolved the Visium slides using five variations of the single-cell reference: the entire dataset, the dataset filtered to nine cell types, and each experimental protocol separately (also filtered to nine cell types). The proportions obtained from using the entire dataset was not used to compute evaluation metrics but only to visualize method stability (*Figure 6—figure supplement 2*). Due to the large size of the reference, we ran some methods differently. We ran stereoscope with 5000 HVGs and subsampled each cell type to a maximum of 250 cells (*-sub*). We gave STRIDE the number of topics to test (*--ntopics 23 33 43 53 63*) instead of the default range that goes up to triple the number of cell types. We ran DestVI with 2500 training epochs and batch size of 128. For Seurat, we ran *FindTransferAnchors* with the 'rpca' (reciprocal PCA) option instead of 'pcaproject'. MuSiC and SpatialDWLS use dense matrices in their code (in contrast to sparse matrices in other methods), resulting in a size limit of $2^{31}$ elements in a matrix. Whenever this is exceeded, we downsampled the reference dataset by keeping maximally 10,000 cells per cell type and keeping 3000 HVGs along with any genes that are at least 10% expressed in a cell type. When a reference with multiple protocols was used, we provided this information to cell2location and Seurat.

To compute the AUPR, we considered only spots annotated as being in the central or portal zone. We considered the following ground truth for portal vein and central vein ECs:

*Continued on next page*

|              | Portal vein EC | Central Vein EC |
|--------------|:--------------:|:---------------:|
| Portal spot  | 1              | 0               |
| Central spot | 0              | 1               |

For the JSD ground truth, we only retained samples from the snRNA-seq protocol where all nine cell types were present, resulting in four samples (ABU11, ABU13, ABU17, and ABU20). For both the ground truth and predicted proportions, we averaged the abundance of each cell type in per slide or sample. Then, we calculated the pairwise JSD between each of the four slides and four snRNA-seq samples and reported the average JSD. The biological variation was obtained by averaging the pairwise JSD between the snRNA-seq samples.

## Melanoma

The scRNA-seq and spatial datasets were downloaded from the original study (*Karras et al., 2022*; *Supplementary file 1d*), with the scRNA-seq dataset being the same one used in the silver standard before the split (*Supplementary file 1b*). We preprocessed the Visium slides by filtering out spots with fewer than 100 features, as their inclusion led to errors when running marker-based deconvolution methods. This filtering removed 26 spots from the third slide. The blood vessel annotation was provided by the pathologist from the original publication. We ran stereoscope and DestVI with the same parameters as the liver case study due to the size of the reference dataset.

The Molecular Cartography (MC) dataset consists of two samples, six sections, and 33 regions of interest (ROI). For each ROI, a DAPI-stained image and a text file containing transcript locations are provided. We obtained cell type proportions using an in-house image processing pipeline. First, the image was cleaned by eliminating tiling effects and background debris using BaSic (*Peng et al., 2017*) and scikit-image (*van der Walt et al., 2014*), respectively. Subsequently, DAPI-stained images were segmented, and locations of individual nuclei were obtained using CellPose (*Stringer et al., 2021*). The transcripts of all measured genes were then assigned to their corresponding cells, resulting in the creation of cell-by-gene count matrices. These matrices were normalized based on the size of segmented nuclei and preprocessed in ScanPy (*Wolf et al., 2018*). Specifically, counts were log-transformed, scaled, and genes expressed in fewer than five cells and cells with less than 10 transcripts were filtered out. Leiden clustering (*Traag et al., 2019*) was performed using 17 principal components and 35 neighbors, and cells were annotated using *scanpy.tl.score_genes* with a curated marker gene list. Finally, the counts of each cell type were aggregated across samples to obtain cell type proportions.

As dendritic cells and melanoma cell states could not be annotated in the MC dataset, we adjusted the predicted proportions from Visium by removing dendritic cells proportions (pDCs and DCs) and recalculating the relative proportions, and aggregating the proportions of the seven malignant cell states.

## Scalability

We generated a synthetic dataset with 10,000 spots and 31,053 genes using only the snRNA-seq protocol from the liver atlas. We then used the remaining two protocols (combined) as the reference dataset. Genes of the synthetic dataset were downsampled randomly based on a uniform distribution, while genes of the reference data were downsampled based on marker genes and HVGs. The spots/cells of both datasets were downsampled randomly, and this was done in a stratified manner for the reference dataset.

## Evaluation metrics and baselines

The root-mean-squared error between the known and predicted proportions ($p$) of a spot $s$ for cell type $z$, in a reference dataset with $Z$ cell types in total, is calculated as

$$RMSE\left(s_{known}, s_{predicted}\right) = \sqrt{\frac{1}{Z}\sum_{z}^{Z}\left(p_{sz,\,known} - p_{sz,\,predicted}\right)^2}$$

We calculated the JSD and AUPR using the R packages *philentropy* and *precrec*, respectively (**Saito and Rehmsmeier, 2017**; **Drost, 2018**). The JSD is a smoothed and symmetric version of the Kullback-Leibler divergence (KL). It is calculated as

$$JSD(s_{known}\|s_{predicted}) = \frac{1}{2}KL(p_{s,known}\|M) + \frac{1}{2}KL(p_{s,predicted}\|M),$$

where

$$M = \frac{1}{2}\left(p_{s,\,known} + p_{s,\,predicted}\right)$$

To calculate the AUPR, we binarized the known proportions by considering a cell type to be present in a spot if its proportion is greater than zero, and absent if it is equal to zero. Then, we compute the micro-averaged AUPR by aggregating all cell types together.

For the silver standards, the RMSE and JSD values across all spots are averaged and used as the representative value for a replicate $k$ for each dataset-abundance pattern combination. In contrast, only one AUPR value is obtained per replicate. Note that the RMSE calculation takes the number of cell types into account and hence should not be compared between datasets.

We created the performance summary plot using the *funkyheatmap* package in R. To aggregate per data source and per abundance pattern, we calculated the grand mean of each metric across datasets, applied min-max scaling on each metric, and then computed the geometric mean of the scaled metrics (RMSE, AUPR, and JSD for gold and silver standards; AUPR and JSD for liver; and only JSD for melanoma). The aggregation per metric is the arithmetic mean of all datasets evaluated on that metric. Finally, we determined the overall rankings based on the weighted ranks of the following criteria: silver standard, gold standard, liver case study, melanoma case study, rare cell type detection, stability, and runtime. We assigned a weight of 0.5 to each silver standard abundance pattern and a weight of one to the remaining criteria.

To provide reference values for each metric, we used (1) random proportions based on probabilities drawn from a Dirichlet distribution and (2) predictions from the NNLS algorithm. For the former, we used the *DirichletReg* package (**Maier, 2014**) to generate reference values for all 63 silver standards using the average value across 100 iterations. The dimension of the $\alpha$ vector was equal to the number of cell types in the corresponding dataset, and all concentration values were set to one. For the NNLS baseline, we used the Lawson-Hanson NNLS implementation from the *nnls* R package. We solve for $\beta$ in the equation $Y = X\beta$, with $Y$ the spatial expression matrix and $X$ the average gene expression profile per cell type from the scRNA-seq reference. We obtained proportion estimates by dividing each element of $\beta$ with its total sum.

## Acknowledgements

We thank all the authors of the methods who provided valuable feedback on how to optimally run their algorithms. Their input has been crucial in enabling us to compare the methods in a fair manner. We thank Lotte Pollaris for processing the Molecular Cartography dataset. CS is funded by the Ghent University Special Research Fund [grant number BOF21-DOC-105], RB is funded by The Research Foundation – Flanders [grant number 1181318 N], RS is funded by the Flemish Government under the Flanders AI Research Program, and YS is funded by Ghent University Special Research Fund [grant number BOF18-GOA-024], and The Research Foundation – Flanders [Excellence of Science (EOS) program and SBO project, grant number S001121N].

# Additional information

## Funding

| Funder | Grant reference number | Author |
|--------|------------------------|--------|
| Bijzonder Onderzoeksfonds UGent | BOF21-DOC-105 | Chananchida Sang-aram |
| Fonds Wetenschappelijk Onderzoek | 1181318N | Robin Browaeys |
| Vlaamse Overheid | Onderzoeksprogramma Artificiele Intelligentie | Ruth Seurinck |
| Fonds Wetenschappelijk Onderzoek | EOS (Excellence of Science) | Yvan Saeys |
| Bijzonder Onderzoeksfonds UGent | BOF18-GOA-024 | Yvan Saeys |
| Fonds Wetenschappelijk Onderzoek | SBO - S001121N | Yvan Saeys |

The funders had no role in study design, data collection and interpretation, or the decision to submit the work for publication.

## Author contributions
Chananchida Sang-aram, Conceptualization, Data curation, Software, Formal analysis, Validation, Investigation, Visualization, Methodology, Writing – original draft, Writing – review and editing; Robin Browaeys, Conceptualization, Data curation, Software, Methodology, Writing – review and editing; Ruth Seurinck, Conceptualization, Supervision, Methodology, Project administration, Writing – review and editing; Yvan Saeys, Conceptualization, Resources, Supervision, Funding acquisition, Project administration, Writing – review and editing

## Author ORCIDs
Chananchida Sang-aram ⓘ https://orcid.org/0000-0002-0922-0822
Robin Browaeys ⓘ https://orcid.org/0000-0003-2934-5195
Yvan Saeys ⓘ https://orcid.org/0000-0002-0415-1506

Reviewer #1 (Public Review): https://doi.org/10.7554/eLife.88431.3.sa1
Reviewer #2 (Public Review): https://doi.org/10.7554/eLife.88431.3.sa2
Author response https://doi.org/10.7554/eLife.88431.3.sa3

---

# Additional files

## Supplementary files
• Supplementary file 1. An overview of all datasets used in this study, including the gold standards, silver standards, liver case study, and melanoma case study.

• Supplementary file 2. An overview of the deconvolution tools benchmarked in this study, including the algorithm and usage information.

• MDAR checklist

## Data availability
All datasets used in this article, including the silver standards, gold standards, and case studies, are available on Zenodo at https://doi.org/10.5281/zenodo.5727613. Original download links and accession numbers of individual studies can be found at *Supplementary file 1*. The synthspot R package can be downloaded from https://github.com/saeyslab/synthspot, copy archived at *Browaeys and Sang-aram, 2024*. The Nextflow pipeline along with analysis scripts can be found at https://github.com/saeyslab/spotless-benchmark, copy archived at *Sang-aram, 2023*.

The following dataset was generated:

| Author(s) | Year | Dataset title | Dataset URL | Database and Identifier |
|---|---|---|---|---|
| Sang-aram C | 2024 | Benchmark datasets for Spotless | https://doi.org/10.5281/zenodo.5727613 | Zenodo, 10.5281/zenodo.5727613 |

The following previously published datasets were used:

| Author(s) | Year | Dataset title | Dataset URL | Database and Identifier |
|---|---|---|---|---|
| Wang X, Allen WE, Wright MA, Sylwestrak EL, Samusik N, Vesuna S, Evans K, Liu C, Ramakrishnan C, Liu J, Nolan GP, Bava FA, Deisseroth K | 2018 | Gold standard - STARMap | https://www.dropbox.com/sh/f7ebheru1lbz91s/AACIAqjvDv--mhnE-PmSXB41a/visual_1020?dl=0 | Dropbox, visual_1020 |
| Tasic B, Yao Z, Graybuck LT, Smith KA, Nguyen TN, Bertagnolli D, Goldy J, Garren E, Economo MN, Viswanathan S, Penn O | 2018 | Shared and distinct transcriptomic cell types across neocortical areas | https://www.ncbi.nlm.nih.gov/geo/query/acc.cgi?acc=GSE115746 | NCBI Gene Expression Omnibus, GSE115746 |
| Saunders A, Macosko EZ, Wysoker A, Goldman M, Krienen FM, de Rivera H, Bien E, Baum M, Bortolin L, Wang S, Goeva A | 2018 | A Single-Cell Atlas of Cell Types, States, and Other Transcriptional Patterns from Nine Regions of the Adult Mouse Brain | https://www.ncbi.nlm.nih.gov/geo/query/acc.cgi?acc=GSE116470 | NCBI Gene Expression Omnibus, GSE116470 |
| Kozareva V, Martin C, Osorno T, Rudolph S, Guo C, Vanderburg C, Nadaf N, Regev A, Regehr WG, Macosko E | 2021 | A transcriptomic atlas of mouse cerebellar cortex reveals novel cell types | https://www.ncbi.nlm.nih.gov/geo/query/acc.cgi?acc=GSE165371 | NCBI Gene Expression Omnibus, GSE165371 |
| Zeisel A, Hochgerner H, Lönnerberg P, Johnsson A, Memic F, Van Der Zwan J, Häring M, Braun E, Borm LE, La Manno G, Codeluppi S | 2018 | Silver standard - mouse hippocampus | http://mousebrain.org/adolescent/downloads.html | Linarsson Lab Mouse Brain Atlas, adolescent |
| Park J, Shrestha R, Qiu C, Kondo A, Huang S, Werth M, Li M, Barasch J, Suszták K | 2018 | Comprehensive single cell RNAseq analysis of the kidney reveals novel cell types and unexpected cell plasticity | https://www.ncbi.nlm.nih.gov/geo/query/acc.cgi?acc=GSE107585 | NCBI Gene Expression Omnibus, GSE107585 |

*Continued on next page*

*Continued*

| Author(s) | Year | Dataset title | Dataset URL | Database and Identifier |
|---|---|---|---|---|
| Karras P, Bordeu I, Pozniak J, Nowosad A, Pazzi C, Van Raemdonck N, Landeloos E, Van Herck Y, Pedri D, Bervoets G, Makhzami S | 2022 | A cellular hierarchy in melanoma uncouples growth and metastasis | https://www.ncbi.nlm.nih.gov/geo/query/acc.cgi?acc=GSE207592 | NCBI Gene Expression Omnibus, GSE207592 |
| Ji AL, Rubin AJ, Thrane K, Jiang S, Reynolds DL, Meyers RM, Guo MG, George BM, Mollbrink A, Bergenstråhle J, Larsson L | 2022 | Single Cell and Spatial Analysis of Human Squamous Cell Carcinoma [single-cell RNA-seq] | https://www.ncbi.nlm.nih.gov/geo/query/acc.cgi?acc=GSE144236 | NCBI Gene Expression Omnibus, GSE144236 |
| Yao Z, Van Velthoven CT, Nguyen TN, Goldy J, Sedeno-Cortes AE, Baftizadeh F, Bertagnolli D, Casper T, Chiang M, Crichton K, Ding SL | 2021 | Stability analysis - mouse brain cortex | https://portal.brain-map.org/atlases-and-data/rnaseq/mouse-whole-cortex-and-hippocampus-10x | Allen Brain Atlas, Mouse-whole-cortex-and-hippocampus-10x |
| Guilliams M, Bonnardel J, Haest B, Vanderborght B, Wagner C, Remmerie A, Bujko A, Martens L, Thoné T, Browaeys R, De Ponti FF | 2022 | Case study - mouse liver | https://www.livercellatlas.org/download.php | Scott Lab & Guilliams Lab Liver Cell Atlas, Mouse-StSt |
| Karras P, Bordeu I, Pozniak J, Nowosad A, Pazzi C, Van Raemdonck N, Landeloos E, Van Herck Y, Pedri D, Bervoets G, Makhzami S | 2022 | Case study - mouse melanoma (10x Visium) | https://drive.google.com/drive/folders/1poq4Lo5AxVp0WpG1EMgIjIeDR4q98zcA | Google Drive, 1poq4Lo5AxVp0WpG1EMgIjIeDR4q98zcA |
| Karras P, Bordeu I, Pozniak J, Nowosad A, Pazzi C, Van Raemdonck N, Landeloos E, Van Herck Y, Pedri D, Bervoets G, Makhzami S | 2022 | A cellular hierarchy in melanoma uncouples growth and metastasis | https://doi.org/10.5281/zenodo.6856193 | Zenodo, 10.5281/zenodo.6856193 |

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

## Appendix 1

### Description and validation of the simulation procedure

To generate synthetic spots, *synthspot* by default samples 2–10 cells from input scRNA-seq data. The counts from these cells are then summed up per gene then downsampled using the *downsampleMatrix* function from *DropletUtils* (*Lun et al., 2019*) to be within the given mean and standard deviation (20,000±5000 counts by default). The uniqueness of *synthspot* lies in the variable cell type frequency priors between abundance patterns, which determine the probability that a cell type will be sampled during spot generation. The nine abundance patterns used in our benchmark are made up of three characteristics, (1) the uniformity of each region, the (2) distinctness of cell types within each region, and (3) whether or not there are missing, dominant, or rare cell types (*Figure 1—figure supplement 1*). *Appendix 1—figure 1* depicts the simulation process in detail. For simplicity purposes, we have excluded the *partially dominant*, *regionally rare*, and *missing cell types* abundance patterns from this flowchart. For the *partially dominant* pattern, we would also randomly select a region the dominant cell type would be absent (cell type prior is then set to zero), and another region where it is as equally abundant as other cell types (prior is sampled from uniform distribution of *Wagner et al., 2016*; *Li et al., 2022*). Similarly, for the *regionally rare* pattern, we would select a select where the rare cell type will only be present in, and then the prior of the rare cell type would be set to zero for all other regions. For the *missing cell types* pattern, we would first randomly select four cell types to be removed. Note that it is also possible for synthspot to use the cell type composition of regionally annotated scRNA-seq data as frequency priors. In that case, it will create the number of regions equal to the scRNA-seq data and use cell type frequencies in the corresponding real region.

As an example, let us say we want to create a synthetic dataset containing two artificial regions which follow the *dominant cell type* pattern. Consider an input scRNA-seq dataset with eight cell types called A, B, C, …, H. In this case, a dominant cell type is randomly selected which will be present in all regions (=H). Assume that region 1 contains cell types [A, B, C, E, H]. The abundance of H is sampled from a uniform distribution from 75 to 100, while the rest will be sampled from 1 to 10. Suppose we obtain the following priors: H=80, A=1, B=2, C=3, and E=4. The summed abundance is 90, and the cell type frequencies are now: H=0.9, A=0.01, B=0.02, C=0.03 and E=0.04. For all spots generated in region 1, these are the probabilities in which the cell types will be sampled.

We validated that our synthetic data and its abundance patterns sufficiently matches real Visium data in two ways, comparing the distributions with *countsimQC* (*Soneson, 2018*) and using frequency priors based on real data. We compared synthspot with the algorithms to generate synthetic data used by cell2location, stereoscope, and SPOTlight. We used brain and kidney Visium datasets as the reference, and generated synthetic data using scRNA-seq data from the respective organs. These were the same scRNA-seq datasets used in our silver standard.

The counts per gene of each dataset seem to be representative of each algorithm's performance for other metrics, for example expression distribution, dispersion, mean-variance trend, and fraction of zeros (*Appendix 1—figures 2 and 3*). For the brain dataset, synthspot has the most resemblance with real data, followed by SPOTlight (*Appendix 1—figure 2*). Since cell2location and stereoscope did not implement a downsampling step in their simulation, their synthetic brain datasets had overly abundant counts, a result of the plate-based scRNA-seq dataset (SMART-seq). SPOTlight downsamples each spot to have a total UMI count of 20,000, so the count distribution becomes uniform, unlike real data. For the kidney, all algorithms except stereoscope's were able to generate synthetic data that resembled real data (*Appendix 1—figure 3*). For most measures, cell2location's algorithm had the most resemblance with real data. Nonetheless, the robustness of synthspot towards sequencing technologies of the input dataset along with the flexibility in fine-tuning properties of the synthetic data makes it the preferred tool to aid in benchmarking. Finally, we verified that the distributions of the nine synthspot abundance patterns did not differ from one another visually or from the *real* abundance pattern, which uses real annotations as the frequency priors (*Appendix 1—figure 4*).

As a second verification, we used the regional annotation in the brain cortex dataset with the *real* abundance pattern to generate synthetic data with five brain regions (L1, L2/3, L4, L5, and L6), with each region having the same composition as the real layer. We then compared method performance between the *real* and *diverse overlap* patterns (*Appendix 1—figure 5*). Although the

spot compositions between the patterns are different, method performances are similar, validating that artificial patterns can be used to evaluate model performance.

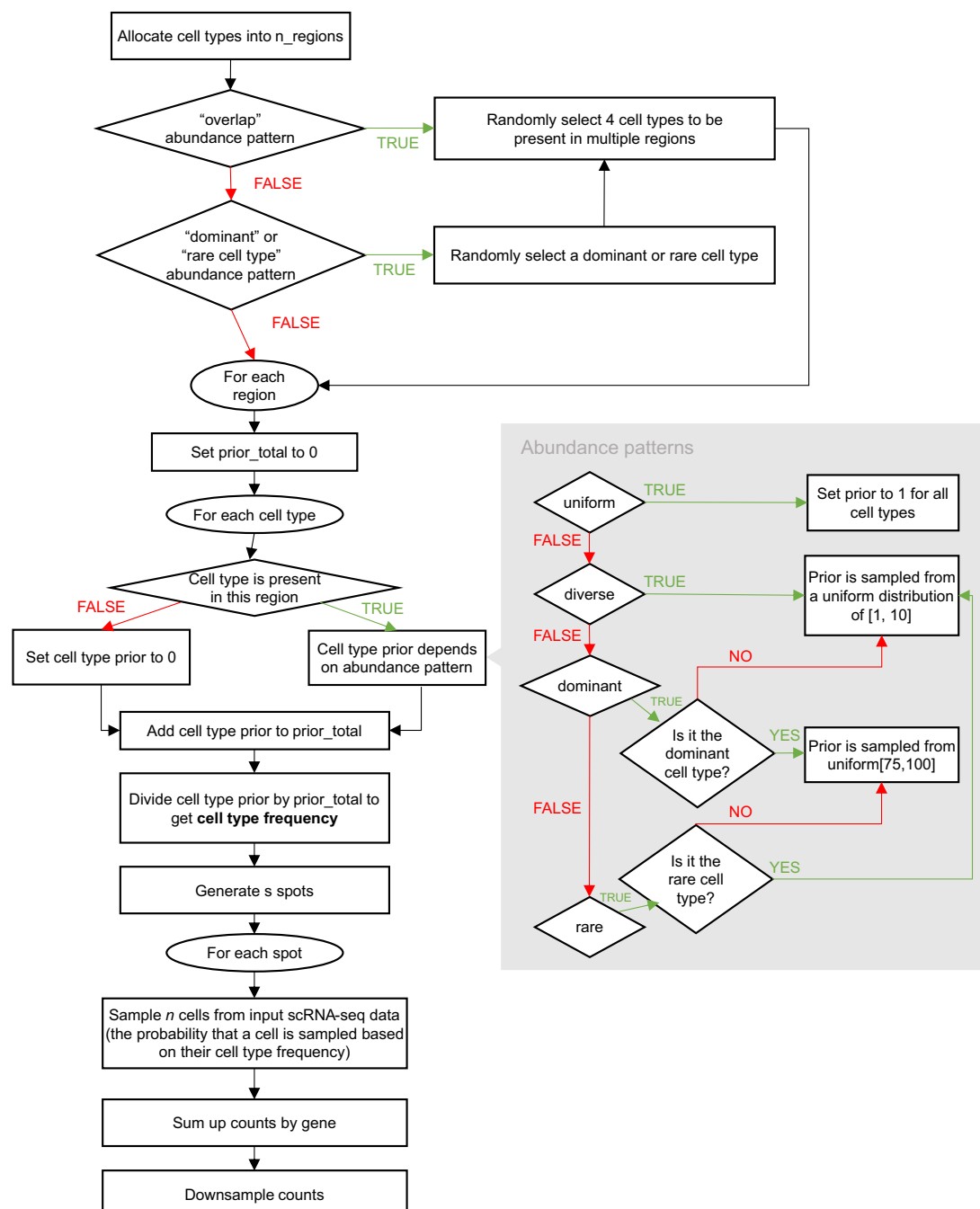

**Appendix 1—figure 1.** Schematic of the *synthspot* simulation algorithm.

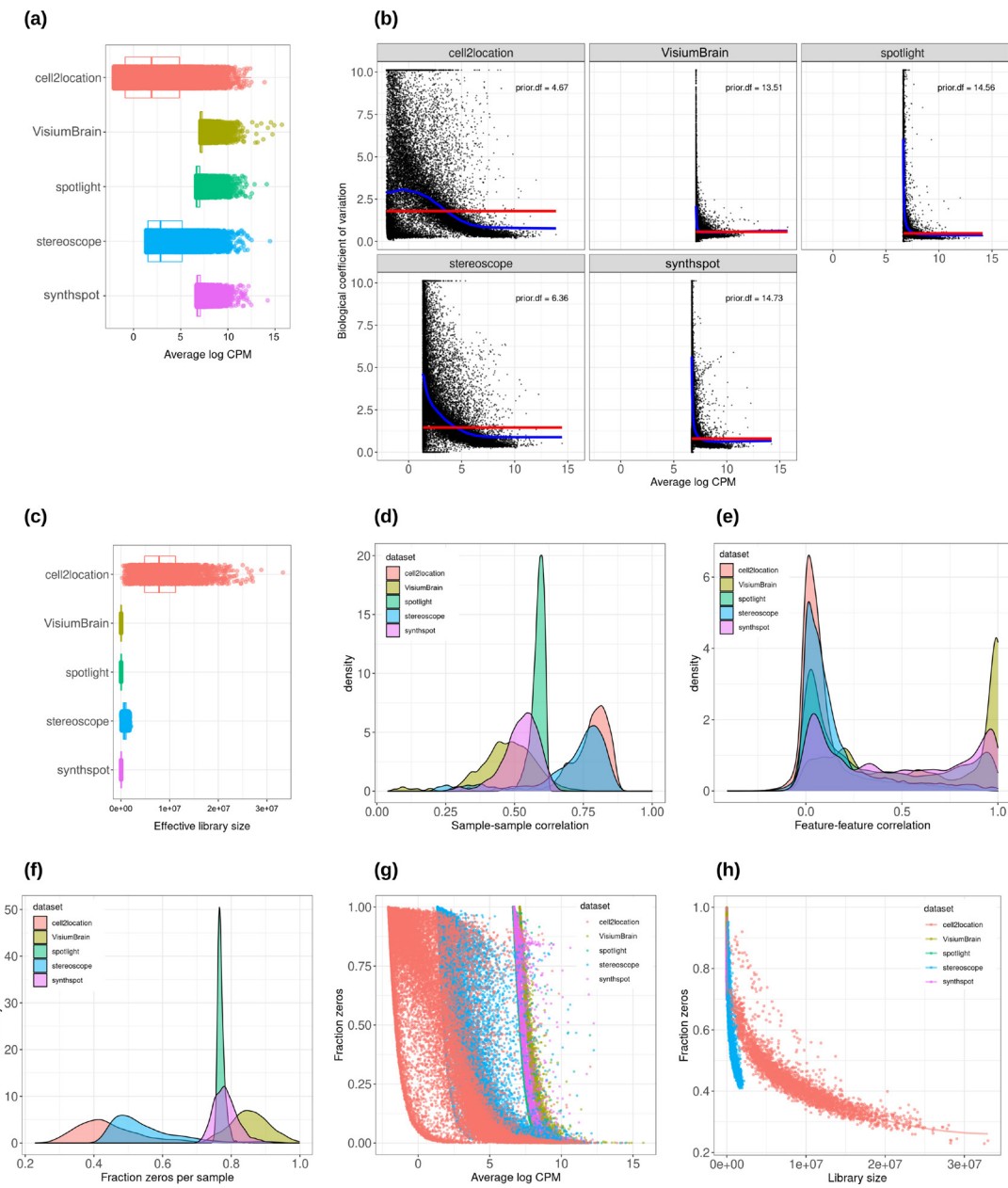

**Appendix 1—figure 2.** Plots comparing the characteristics of real Visium data from mouse brain and synthetic datasets generated from brain scRNA-seq data using different algorithms. (**a**) Average abundance values (log counts per million) per gene. (**b**) Association between average abundance and the dispersion. (**c**) Distribution of effective library sizes, or the total count per sample multiplied by the corresponding TMM normalization factor calculated by *edgeR*. (**d–e**) Distribution of pairwise Spearman correlation coefficients for 500 randomly selected spots (**d**) and genes (**e**), calculated from log CPM values. Only non-constant genes are considered. (**f**) Distribution of the fraction of zeros observed per spot. (**g–h**) The association between fraction zeros and average gene abundance (**g**) and total counts per spot (**h**).

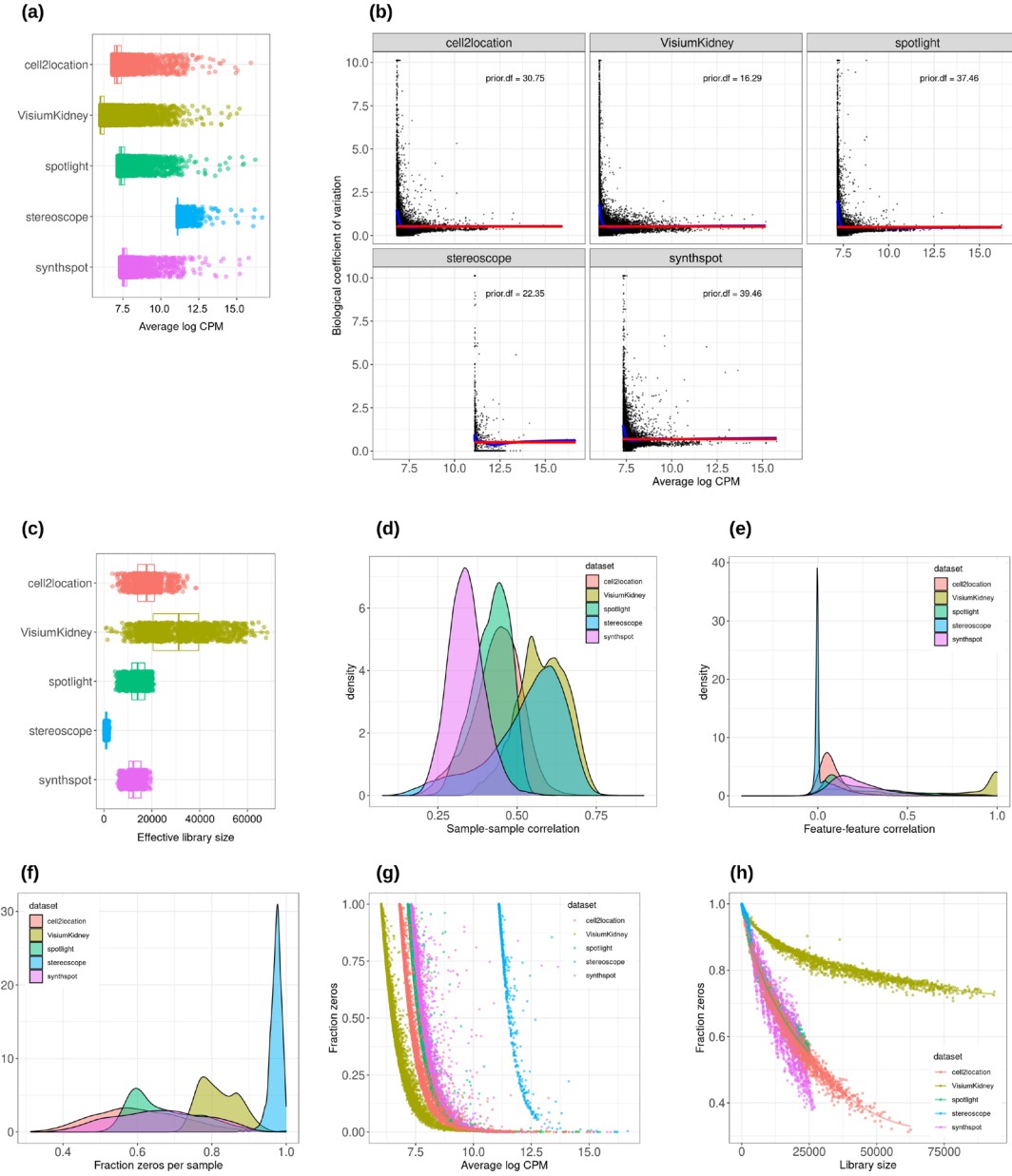

**Appendix 1—figure 3.** Plots comparing the characteristics of real Visium data from mouse kidney and synthetic datasets generated from kidney scRNA-seq data using different algorithms. (**a**) Average abundance values (log counts per million) per gene. (**b**) Association between average abundance and the dispersion. (**c**) Distribution of effective library sizes, or the total count per sample multiplied by the corresponding TMM normalization factor calculated by *edgeR*. (**d–e**) Distribution of pairwise Spearman correlation coefficients for 500 randomly selected spots (**d**) and genes (**e**), calculated from log CPM values. Only non-constant genes are considered. (**f**) Distribution of the fraction of zeros observed per spot. (**g–h**) The association between fraction zeros and average gene abundance (**g**) and total counts per spot (**h**).

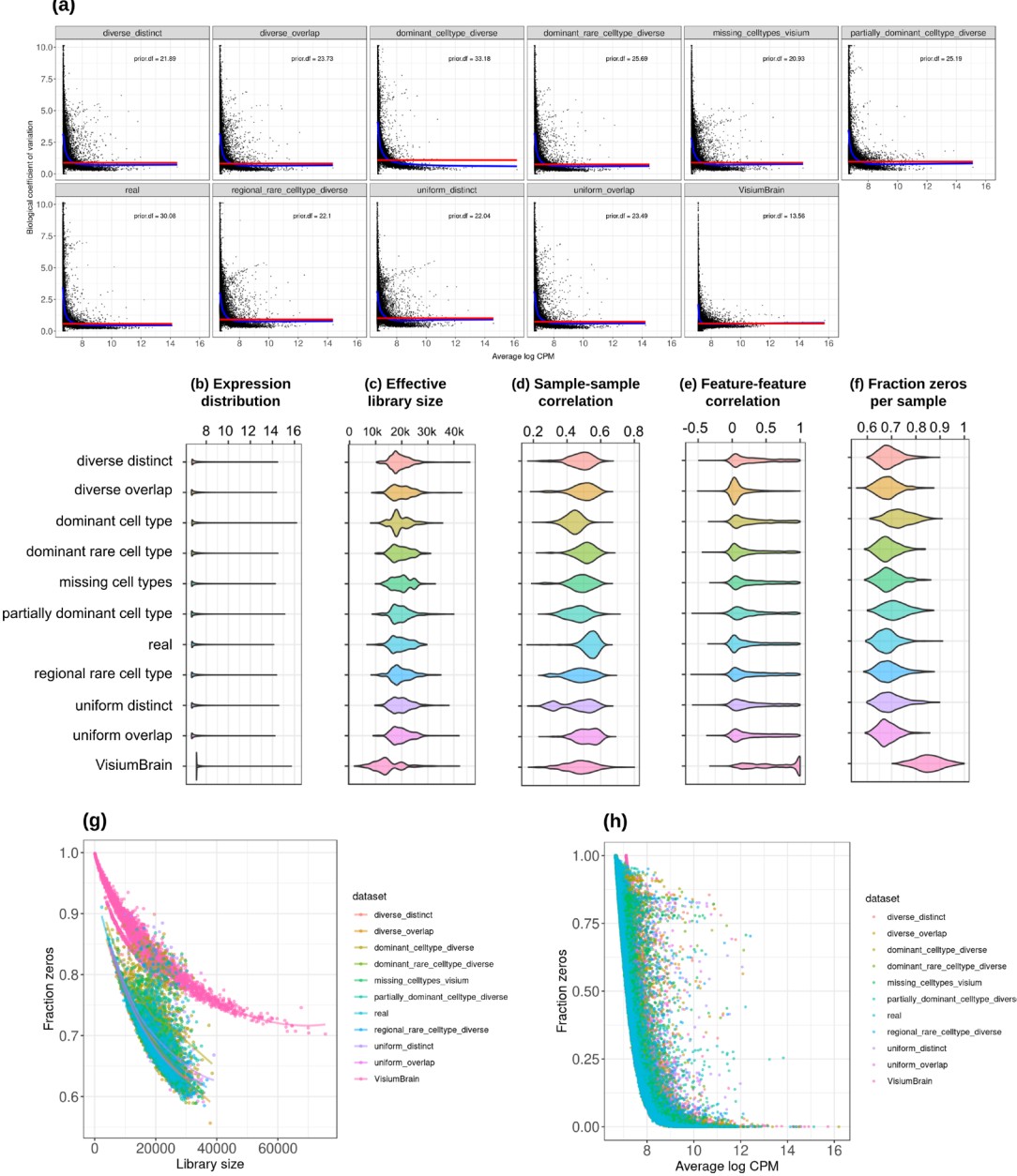

**Appendix 1—figure 4.** Plots comparing the characteristics of real Visium data from mouse brain and the eight synthetic abundance patterns from synthspot generated from brain scRNA-seq data. (**a**) Association between average abundance and the dispersion. (**b**) Average abundance values (log counts per million) per gene. (**c**) Distribution of effective library sizes, or the total count per sample multiplied by the corresponding TMM normalization factor calculated by *edgeR*. (**d–e**) Distribution of pairwise Spearman correlation coefficients for 500 randomly selected spots (**d**) and genes (**e**), calculated from log CPM values. Only non-constant genes are considered. (**f**) Distribution of the fraction of zeros observed per spot. (**g–h**) The association between fraction zeros and average gene abundance (**g**) and total counts per spot (**h**).

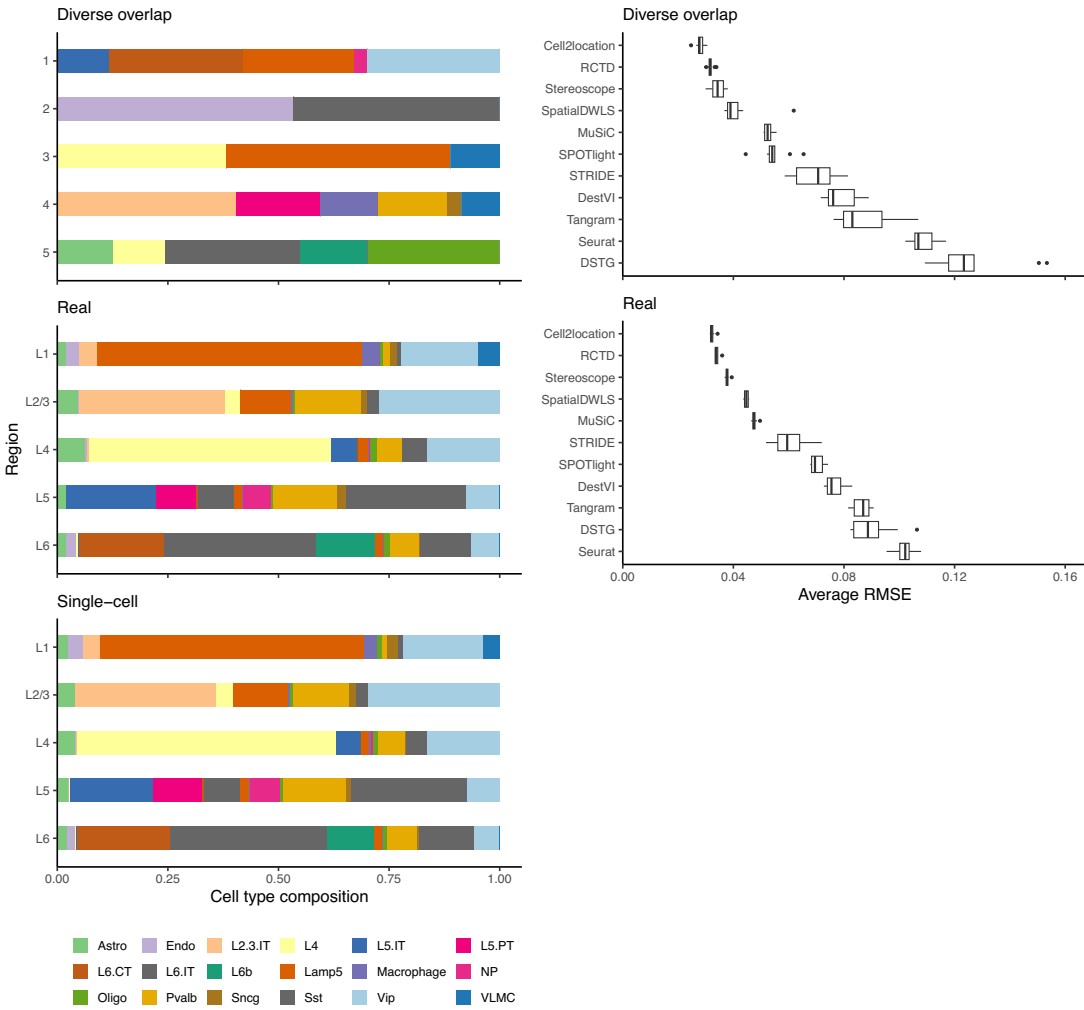

**Appendix 1—figure 5.** Method performance on synthetic data generated from completely synthetic or annotated brain regions. When using an *artificial* abundance pattern (*diverse overlap*) to create synthetic spatial data, method rankings remain almost identical as when using a *real* abundance pattern. The *real* pattern uses regional annotations from the scRNA-seq input to create regions with the same cell type frequencies.

## Appendix 2

### Issues with threshold-based classification metrics

In addition to the area under the precision-recall curve (AUPR), we initially included the (balanced) accuracy, specificity, recall (sensitivity), precision, and F1 score in the evaluation. These metrics evaluate the classification capabilities of each method, that is the correctness of cell type presence and absence prediction. Briefly, accuracy is the percentage of correctly classified cell types, specificity measures how many of the cell types predicted as absent are truly absent, sensitivity measures how well a method can detect a cell type within a spot, precision measures how many cell types predicted as present are truly present, and the F1 score integrates sensitivity and precision.

The first issue was that methods that use probabilistic models (e.g. cell2location, stereoscope, RCTD and DestVI) do not return proportions that are exactly zero but instead negligible values as low as $10^{-9}$. This made an unbiased evaluation difficult since a fixed threshold for cell type presence/absence must be selected to calculate classification metrics. In particular, different methods, datasets and abundance patterns have different thresholds for which the classification metrics are at a maximum.

The second issue stems from the class imbalance of our datasets, in which more cell types are absent than present in a spot (more negative than positive classes). In general, around 15% of the proportion matrix are positives classes, which made the specificity and precision particularly uninformative. This can be seen by how Seurat was the best performer on both specificity and precision despite having low sensitivity (*Appendix 2—figure 1*). By mostly predicting cell types to be absent in a spot, there are more false negatives (FNs), but specificity and precision do not take FNs into account. The balanced accuracy and F1 score were also unable to entirely correct for this class imbalance.

Given these two issues, we decided to use the precision-recall curve (PR) for evaluation instead and not include these five classification metrics (although we discuss its calculation below). The PR curve plots precision against recall at different thresholds, and the threshold to distinguish cell type absence/presence is varied from zero to one instead of fixing it at a certain value. Hence, the proportions are used as-is without rounding or binarization. It is also recommended for use with imbalanced datasets (*Davis, 2006*).

### Calculation of threshold-based classification metrics

First, we rounded the predicted proportion matrices to two decimal points, so a proportion of 0.005 and under was rounded to zero. We calculated the micro-average of each classification metric. This is a global metric where the contributions of all classes are considered. We essentially treat the proportion matrix as in a binary classification problem and go through each element individually. As an example, the micro-precision is calculated as

$$Precision_{micro} = \frac{\sum_z^Z TP_z}{\sum_z^Z TP_z + \sum_z^Z FP_z}$$

where TP stands for true positive and FP for false positive. This is in contrast to the macro-average, where the metric is computed independently for each cell type using a one-vs-all approach, and the average is taken across all cell types ($Precision_{macro} = \frac{1}{Z} \sum_z^Z Precision_z$).

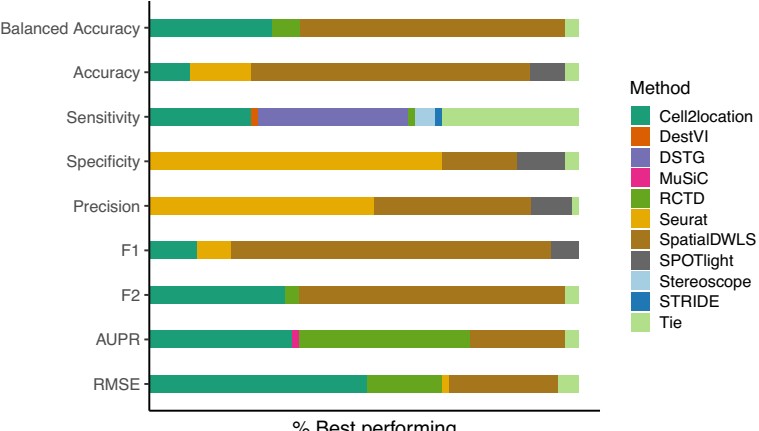

**Appendix 2—figure 1.** The relative frequency in which a method performs best in the silver standard, based on the best median value across ten replicates for that combination. 'Tie' means that two or more methods score the same up to the third decimal point. RMSE: root-mean-square error; AUPR: area under the precision-recall curve.

## Appendix 3

## Method execution and parameter choice

In this appendix, we briefly describe each method, how we ran them, and the parameters that we evaluated. As most methods contain adjustable parameters that can affect its performance, we tested a range of parameter options to ensure optimal performance for each method. For reproducibility, users can find the exact parameters we have used for each analysis under the 'conf/' folder in our Github repository.

### cell2location (v0.06a)

Cell2location models the transcripts with a negative binomial distribution. It uses variational inference to estimate all the parameters. One advantage of cell2location is that users can shape model priors by providing hyperparameters that correspond to their prior knowledge. We used default priors from the tutorial of cell2location version 0.3. We filtered genes from the reference and spatial datasets as suggested by the tutorial. Model fitting was performed with cross-validation stratified by cell type annotation (*stratify_cv*). Sample information was not given to the model. The number of training iterations was set to 30,000 (*n_iter*).

In a newer tutorial, the default value of *detection_alpha* hyperparameter changed from 200 to 20. We compared the old and new values on the brain cortex seqFISH+ dataset (gold standard) and the kidney dataset (silver standard) and did not find a difference in performance (*Appendix 3—figure 1a*). Hence, we kept *detection_alpha = 200* for our benchmark. We also varied the number of cells per location from 10 to 50 cells in the gold standard but also did not find a noticeable change in performance (*Appendix 3—figure 1b*).

### DestVI (v0.16.0)

DestVI uses latent variable models (i.e. variational autoencoders) for the single-cell and spatial data. Unlike other methods, it also models a continuous estimate of cell state for every cell type in every spot. With the default of 2500 epochs, training did not converge for many of the silver standard datasets. We followed the author's recommendations and increased to 5000 training epochs and reduce the minibatch size (*batch_size*) to 64, which improved performance for all silver standard datasets (*Appendix 3—figure 3*). For the gold standards, we used 2500 training epochs and *batch_size = 4*.

### DSTG (v0.0.1)

DSTG uses a graph convolutional neural network to learn the composition of real spots from simulated spots. It performs a joint dimensionality reduction (canonical correlation analysis, CCA) and identifies the mutual nearest neighbors between the real and simulated spots. As many of the parameters were hardcoded, for example, 200 nearest neighbors and 30 canonical vectors, the algorithm did not run on our gold standards where there were only nine spots per FOV. We adjusted the source code to change *k_filter*, *k*, *num_cc*, and *dims* to be equal to the number of spots in such cases, otherwise the default parameters were used.

### MuSiC (v0.2.0)

MuSiC is a bulk deconvolution method developed to handle multiple scRNA-seq references from multiple subjects. It employs weighted non-negative least squares (NNLS) regression. In case there are scRNA-seq reference datasets from multiple samples, the between-subject variance is used as weights for each gene. Genes with consistent expression among subjects are considered more informative and will be given higher weights during regression. We ran MuSiC without pre-grouping of cell types. We did not provide subject information to the model but instead considered each cell as a separate subject. This is not all our silver standards contained sample information, and we observed that the performance remained the same or worse when the sample information was given (*Appendix 3—figure 2a*). We also experimented with creating 'pseudosamples' for datasets without sample information, where each cell was randomly assigned to one out of three artificial samples. This returned a worse performance than when single cells were used as samples (*Appendix 3—figure 2b*).

## RCTD (v1.2.0)

RCTD models the transcripts as being Poisson distributed and uses maximum likelihood estimation to infer cell type proportions. We ran RCTD with *doublet_mode="full"*, indicating that many cell types per spot were to be expected.

## Seurat integration (v4.1.0)

Using joint dimensionality reduction between the scRNA-seq (reference) and spatial (query) data, we defined compatible reference-query pairs and used them as anchors for the label transfer procedure. We obtain a probability for a cell type being in the spot and use this as proxy for the abundance. We compared two normalization methods (SCTransform and vst) and two projection methods (PCA and CCA) and found that using SCTransform with PCA gave the best results (*Appendix 3—figure 4*). As with DSTG, we changed the number of neighbors and vectors (*k. score*, *k.weight*, *dims*) used in the gold standard as equal to the number of spots, otherwise the default was used.

## SpatialDWLS (v1.1.0)

SpatialDWLS performs cell-type enrichment analysis for each spot, then uses down-weighted least squares on marker genes. We followed the protocol described in *Rossi and Chen, 2022*, where the *makeSignMatrixDWLS* function was used with top 100 marker genes, instead of following the online vignette where the *makeSignMatrixPAGE* function was used with all marker genes (*Appendix 3—figure 5*).

## SPOTlight (v0.1.7)

SPOTlight is the only method based on non-negative matrix factorization (NMF) and NNLS. This method computes topics from the gene expression profile with NMF instead of using the expression values directly. NNLS is used with the topics to obtain the cell type proportions for each spot. We followed the vignette of version 0.1.5, as the recommended parameters have changed slightly between each version of SPOTlight. In this version, SPOTlight uses Seurat's *FindAllMarkers* functions to calculate marker genes between cell types, and the *logfc.threshold* (limit testing to genes which show at least X-fold difference) and *min.pct* (only test genes that are expressed in at least X% of cells) parameters have a huge impact on the resulting list of marker genes. Furthermore, the parameters *cl_n* (number of cell types to use) and *min_cont* (only keep cells that are at least X% present in a spot) within the deconvolution function itself affects the predictions. We tested three sets of parameters and in the end, ran *FindAllMarkers* with *only.pos=TRUE*, *logfc.threshold=1*, and *min.pct=0.9*, and the deconvolution with *cl_n=50* and *min_cont = 0.09* (*Appendix 3—figure 6*). This was also the parameter set that gave the shortest runtime. We also normalized both the reference and spatial data with *SCTransform*.

## Stereoscope (v0.2.0)

Stereoscope models the transcripts with a negative binomial distribution. It uses maximum likelihood estimation to infer the rate and overdispersion parameters from the scRNA-seq data and then uses maximum a posterior (MAP) estimation to infer cell type proportions. We ran stereoscope using all genes for the silver standard but only the top 5000 most highly expressed genes for the gold standard, as that gave the best results for both cases. Although the authors have noted in their paper that choosing the 5000 most expressed genes is sufficient, we saw that using all genes for the silver standards still gave slightly better performance (*Appendix 3—figure 7a*). We implemented an option to use the HVGs instead of top expressed genes, but this did not consistently result in better performance (*Appendix 3—figure 7b*). Finally, we tested the *sub* parameter, which subsamples each cell type in the single-cell reference to at most X cells, but did not see any improvement in the kidney (silver standard) or liver dataset (*Appendix 3—figure 7c*). We verified that both the training and test models have converged.

## STRIDE (v0.0.2)

STRIDE trains a topic model from scRNA-seq data, then applies this model to the spatial data. We ran it with `--normalize` and we let STRIDE automatically select the optimal topic number for each dataset. We did not find a pattern when comparing results from raw and normalized counts (*Appendix 3—figure 8*).

## Tangram (v1.0.3)

Tangram learns a spatial alignment of scRNA-seq from the spatial data via nonconvex optimization. Like Seurat integration, Tangram returns the probabilistic counts for each cell type in each cell voxel. We tested the three mapping modes: *cells* (maps single cells to spots), *clusters* (maps average of cells per cell type instead of single cells), and *constrained* (constrain the number of mapped single cell profiles). Although the *constrained* mode was recommended for deconvolution, we found that using the *clusters* resulted in the best performance (*Appendix 3—figure 9*). When running the *constrained* mode, we also provided the ground truth number of cells per spot and total number of cells in the *density_prior* and *target_counts* parameters, respectively. Increasing the training epochs did not have an effect on the performance, and we verified that the models have converged. The final parameters we used were *map_cells_to_space* with *mode="clusters"* and *density_prior="rna_count_based"*. We used the top 100 marker genes for each cell type.

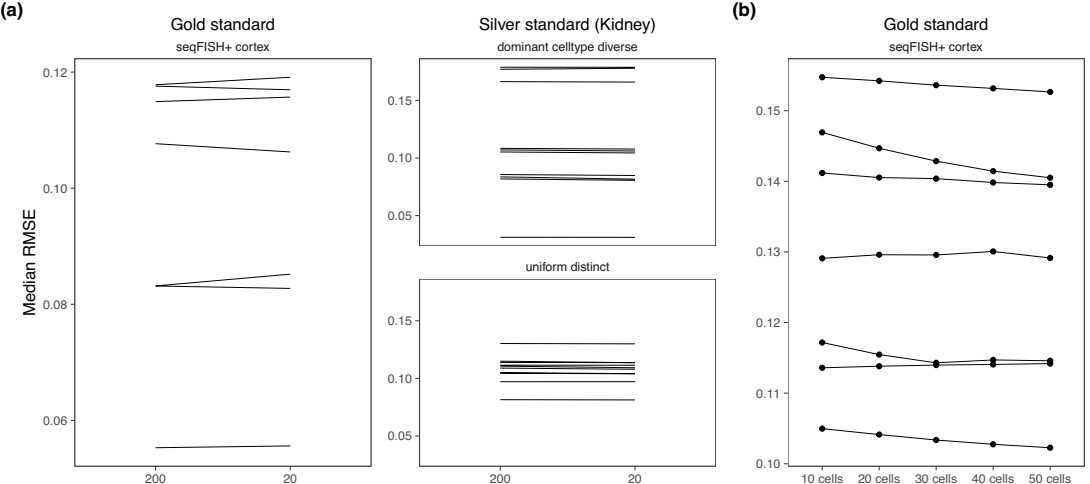

**Appendix 3—figure 1.** Changing hyperparameters in the cell2location model. There is almost no performance difference when changing the (**a**) detection alpha and (**b**) number of cells per spot.

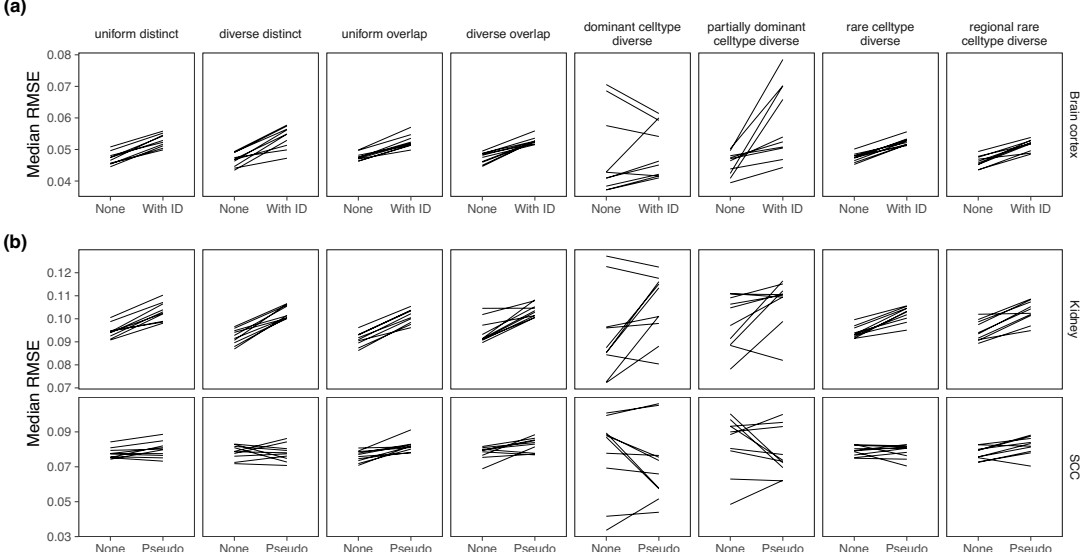

**Appendix 3—figure 2.** Comparing MuSiC performance when the model was given sample information. MuSiC seems to perform best when single cells were used as samples ('None'), as compared to (**a**) when the real sample information was given and (**b**) when pseudosamples were created.

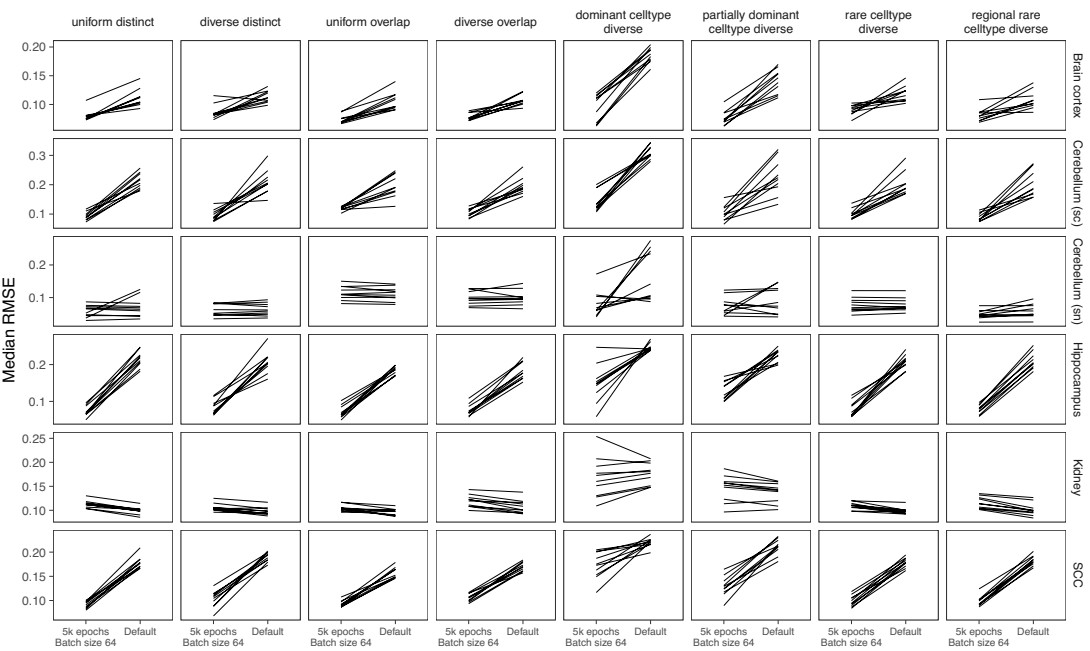

**Appendix 3—figure 3.** Comparing parameters of DestVI. Compared to the default parameters (2500 epochs), DestVI has better performance with 5000 training epochs and batch size of 64.

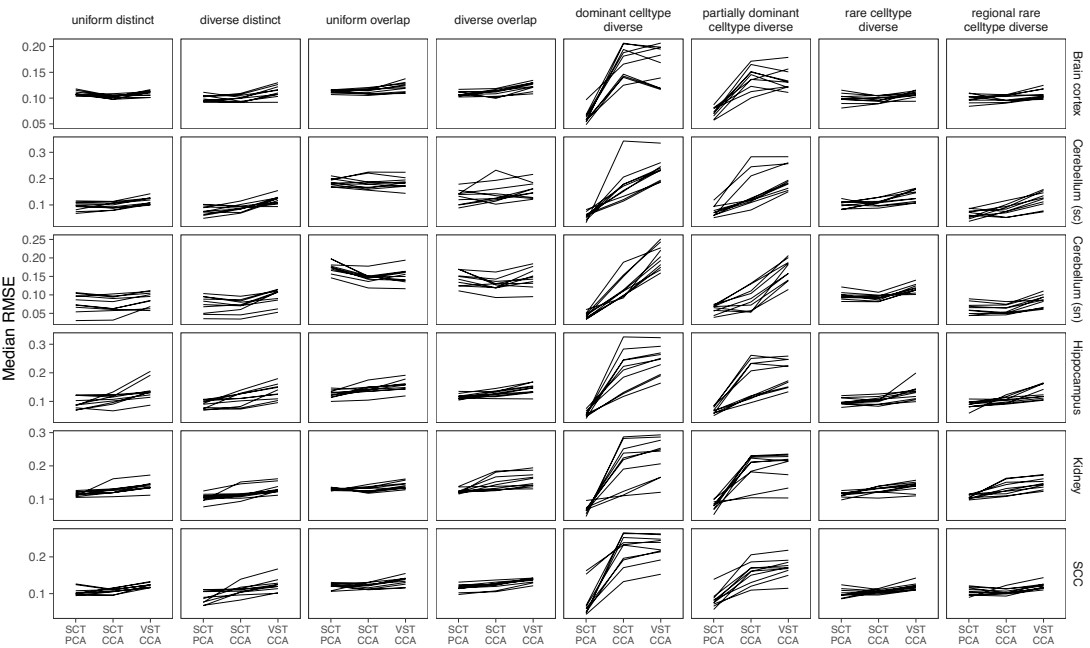

**Appendix 3—figure 4.** Comparing different data transformation and dimensionality reduction methods in Seurat. Seurat has the best performance when the data was transformed using SCTransform and the dimensionality reduction method is PCA. CCA = canonical correlation analysis; PCA = principal component analysis; VST = variance stabilizing transformation.

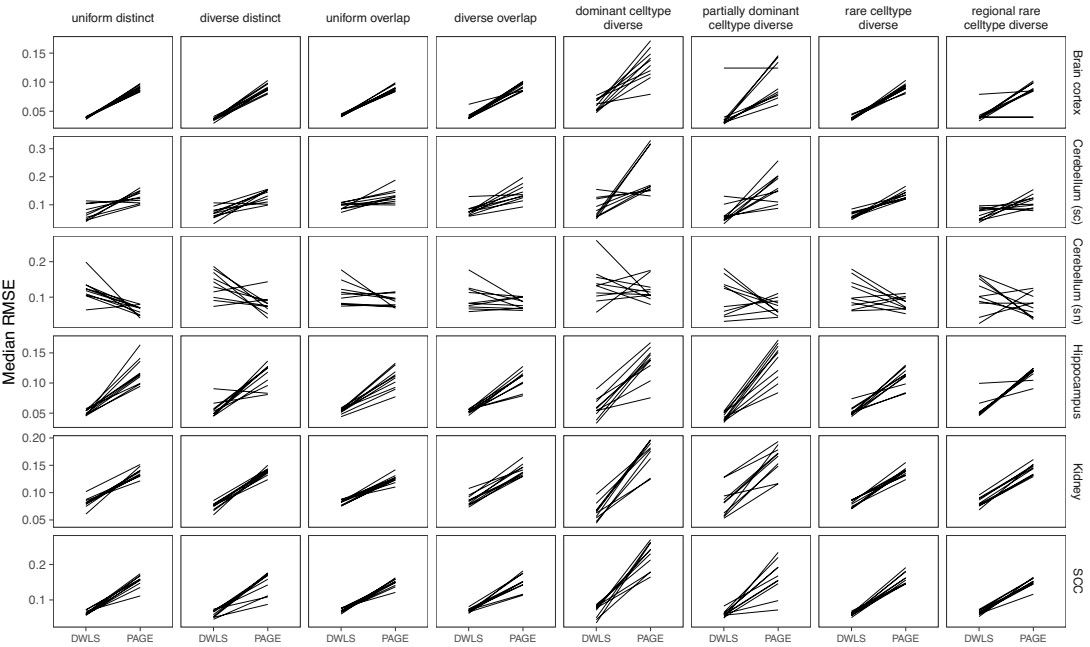

**Appendix 3—figure 5.** Comparing SpatialDWLS between two signature matrix creation functions. SpatialDWLS has better performance when the *makeSignMatrixDWLS* was used to create the signature matrix (as described in the Current Protocols paper), instead of the *makeSignMatrixPAGE* function (described in the online vignette).

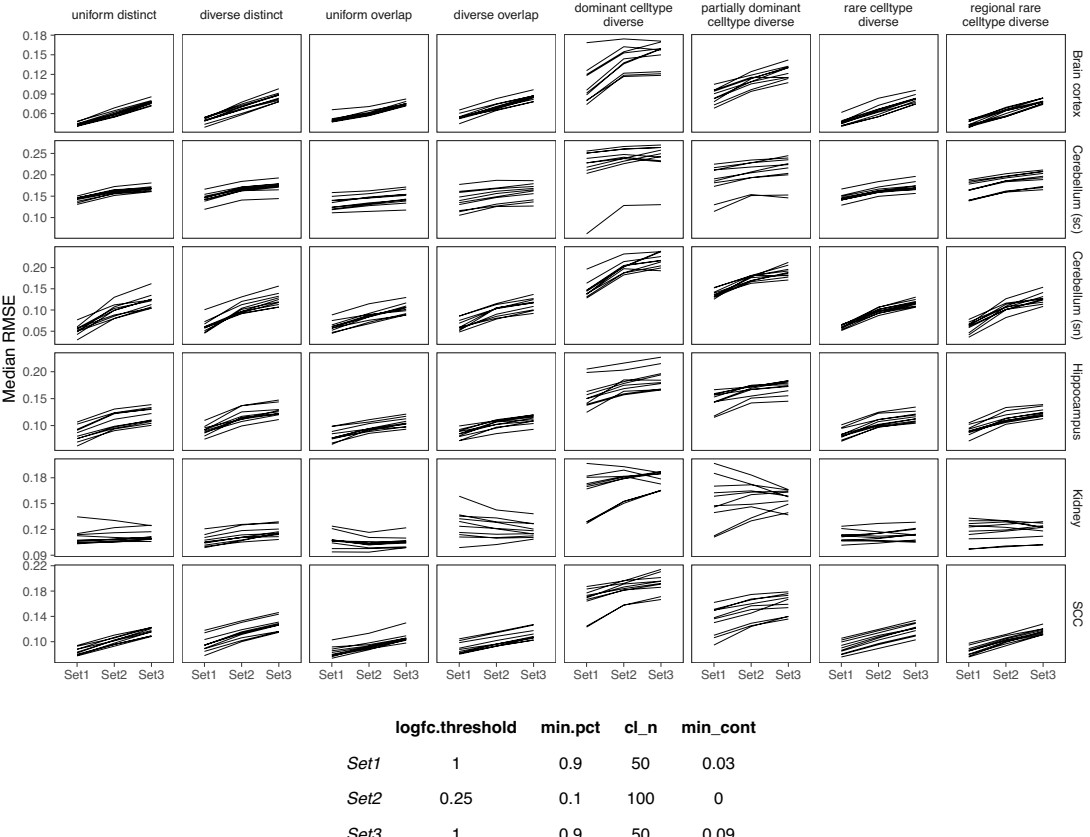

| | logfc.threshold | min.pct | cl_n | min_cont |
|---|---|---|---|---|
| *Set1* | 1 | 0.9 | 50 | 0.03 |
| *Set2* | 0.25 | 0.1 | 100 | 0 |
| *Set3* | 1 | 0.9 | 50 | 0.09 |

**Appendix 3—figure 6.** Comparing SPOTlight performance on three sets of parameters. We used Set1 parameters in our benchmark.

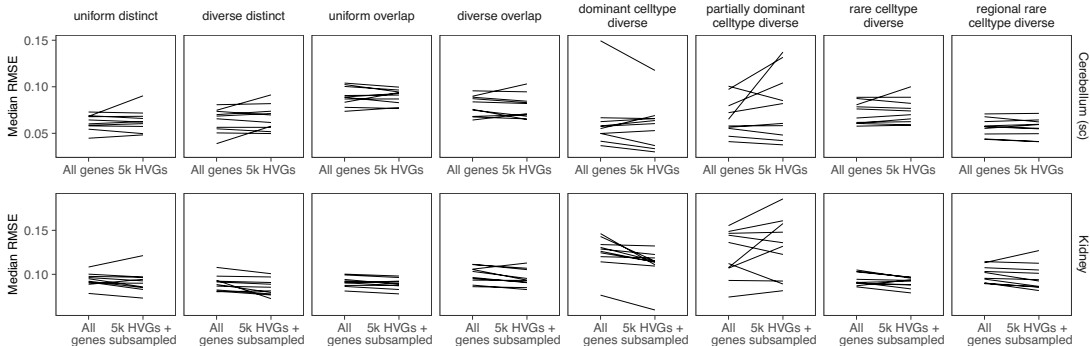

**Appendix 3—figure 7.** Comparing stereoscope performance to using only highly variable genes (HVGs). There is no consistent performance difference between using all genes of stereoscope and using the 5000 HVGs (with or without subsampling the scRNA-seq reference).

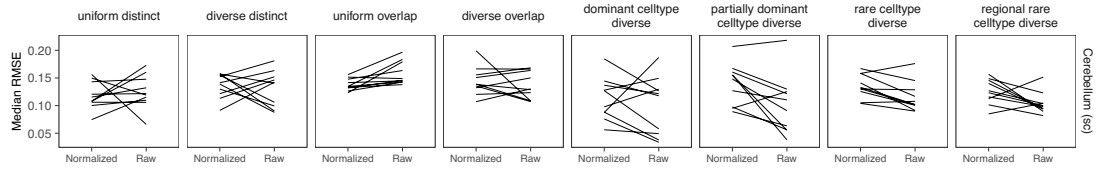

**Appendix 3—figure 8.** Comparing STRIDE performance on normalized and raw counts. For STRIDE, there is no consistent performance difference between normalizing or using the raw counts.

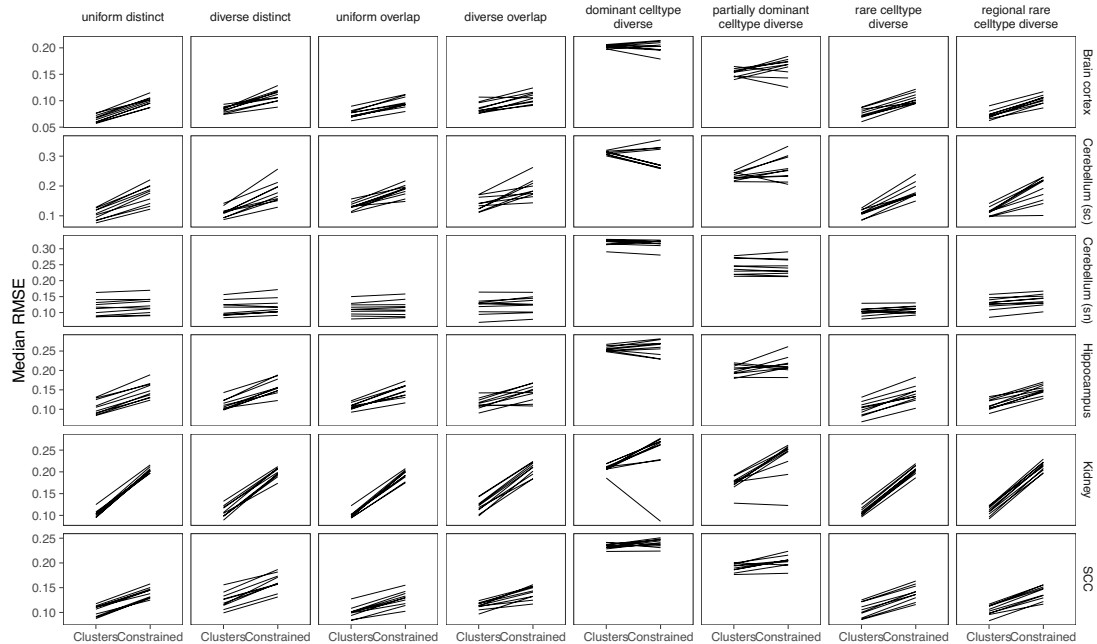

**Appendix 3—figure 9.** Comparing the mapping modes in Tangram. Although the *constrained* mapping mode was recommended in the Tangram vignette, we found that the *clusters* mode achieve better performance.

