## [Editor Report · eLife assessment]

This study makes a **valuable** contribution to spatial transcriptomics by rigorously benchmarking cell-type deconvolution methods, assessing their performance across diverse datasets with a focus on biologically relevant, previously unconsidered aspects. The authors demonstrate the strengths of RCTD, cell2location, and SpatialDWLS for their performance, while also revealing the limitations of many methods when compared to simpler baselines. By implementing a full Nextflow pipeline, Docker containers, and a rigorous assessment of the simulator, this work offers robust insights that elevate the standards for future evaluations and provides a resource for those seeking to improve or develop new deconvolution methods. The thorough comparison and analysis of methods, coupled with a strong emphasis on reproducibility, provide **solid** support for the findings.

---

## [Referee Report · Reviewer #1 (Public Review)]

Cell type deconvolution is one of the early and critical steps in the analysis and integration of spatial omic and single cell gene expression datasets, and there are already many approaches proposed for the analysis. Sang-aram et al. provide an up-to-date benchmark of computational methods for cell type deconvolution.

In doing so, they provide some (perhaps subtle) additional elements that I would say are above the average for a benchmarking study: (i) a full Nextflow pipeline to reproduce their analyses; (ii) methods implemented in Docker containers (which can be used by others to run their datasets); (iii) a fairly decent assessment of their simulator compared to other spatial omics simulators. A key aspect of their results is that they are generally very concordant between real and synthetic datasets. And, it is important that they authors include an appropriate "simpler" baseline method to compare against and surprisingly, several methods performed below this baseline. Overall, this study also has the potential to also set the standard of benchmarks higher, because of these mentioned elements.

The only weakness of this study that I can readily see is that this is a very active area of research and we may see other types of data start to dominate (CosMx, Xenium) and new computational approaches will surely arrive. The Nextflow pipeline will make the the prospect of including new reference datasets and new computational methods easier.

---

## [Referee Report · Reviewer #2 (Public Review)]

In this manuscript Sangaram et al provide a systematic methodology and pipeline for benchmarking cell type deconvolution algorithms for spatial transcriptomic data analysis in a reproducible manner. They developed a tissue pattern simulator that starts from single-cell RNA-seq data to create silver standards and used spatial aggregation strategies from real in situ-based spatial technologies to obtain gold standards. By using several established metrics combined with different deconvolution challenges they systematically scored and ranked 12 different methods and assessed both functional and usability criteria. Altogether, they present a reusable and extendable platform and reach very similar conclusions to other deconvolution benchmarking paper, including that RCTD, SpatialDWLS and Cell2location typically provide the best results. Major strengths of the simulation engine include the ability to downsample and recapitulate several cell and tissue organization patterns.

More specifically, the authors of this study sought to construct a methodology for benchmarking cell type deconvolution algorithms for spatial transcriptomic data analysis in a reproducible manner. The authors leveraged publicly available scRNA-seq, seqFISH, and STARMap datasets to create synthetic spatial datasets modeled after that of the Visium platform. It should be noted that the underlying experimental techniques of seqFISH and STARMap (in situ hybridization) do not parallel that of Visium (sequencing), which could potentially bias simulated data. Furthermore, to generate the ground truth datasets cells and their corresponding count matrix are represented by simple centroids. Although this simplifies the analysis it might not necessarily accurately reflect Visium spots where cells could lie on a boundary and affect deconvolution results.

The authors thoroughly and rigorously compare methods while addressing situational discrepancies in model performance, indicative of a strong analysis. The authors make a point to address both inter- and intra- dataset reference handling, which has a significant impact on performance, as the authors note in the text and conclusions. Indeed, supplying optimal reference data is - potentially most - important to achieve the best performance and hence it's important to understand that experimental design or sample matching is at least equally important to selecting the ideal deconvolution tool.

Similarly, the authors conclude that many methods are still outperformed by bulk deconvolution methods (e.g. Music or NNLS), however, it needs to be noted that these 'bulk' methods are also among the most sensitive when using an external (inter) dataset (S10), which likely resembles the more realistic scenario for most labs.

As the authors also discuss it's important to realize that deconvolution approaches are typically part of larger exploratory data analysis (EDA) efforts and require users to change parameters and input data multiple times. Thus, running time, computing needs, and scalability are probably key factors that researchers would like to consider when looking to deconvolve their datasets.

The authors achieve their aim to benchmark different deconvolution methods and the results from their thorough analysis support the conclusions that creating cell type deconvolution algorithms that can handle both cell abundance and rarity throughout a given tissue sample are challenging.

The reproducibility of the methods described will have significant utility for researchers looking to develop cell type deconvolution algorithms, as this platform will allow simultaneous replication of the described analysis and comparison to new methods.

---

## [Author Response]

The following is the authors’ response to the original reviews.

**Reviewer #1 (Recommendation for the authors)**
I only have one comment for improvement of this study and it has to do with the comparison of simulators that they conducted. There are many other simulators around now, including scDesign3, spaSim, SPIDER, SRTSIM, etc. Are any of those methods worth including in the comparison?

Indeed, many of the mentioned simulators did not exist when we initially developed synthspot, and upon closer examination, they are not directly comparable to our tool.

• scDesign3: The runtime of scDesign3 is quite long as a result of its generative model. The example provided in its tutorial only simulates 183 genes and takes over seven minutes when using four cores on a system with Intel Xeon E5-2640 CPUs running at 2.5GHz. In a small downsampling analysis, we simulated 10, 50, 100, and 150 genes with scDesign3 and observed runtimes of 30, 130, 245, and 360 seconds, respectively. This seems to indicate a linear relationship between the number of genes and the runtime, therefore rendering it unsuitable for simulating whole-transcriptome datasets for deconvolution.

• spaSim: spaSim focuses on modelling cell locations in different tissue structures but does not provide gene expression data. It is designed for testing cell colocalization capabilities rather than simulating gene expression.

• SPIDER: Although SPIDER appears to have some overlap with our work, it seems to be in the early stages of development. The GitHub repository contains only two scripts without any documentation, and the preprint does not provide instructions on how to use the tool.

• SRTSim: SRTSim explicitly states in its publication that it is not suitable for evaluating cell type deconvolution, as its focus is on simulating gene expression data without modelling cell type composition.

• scMultiSim: scMultiSim, like scDesign3, is limited in its capability to model the entire transcriptome.

Nonetheless, the inherent modularity of our Nextflow framework makes it possible for users to simply run the deconvolution methods on data that has been simulated by other simulators if need be.

Additionally, we have added the following rationale for why we developed synthspot in “Synthspot allows simulation of artificial tissue patterns”:

“On the other hand, general-purpose simulators are typically more focused on other inference tasks, such as spatial clustering and cell-cell communication, and are unsuitable for deconvolution. For instance, generative models and kinetic models like those of scDesign3 and scMultiSim are computationally intensive and unable to model entire transcriptomes. SRTSim focuses on modeling gene expression trends and does not explicitly model tissue composition, while spaSim only models tissue composition without gene expression.”

The other aspect of the simulation comparison that I'm missing is some kind of spatial metric. There are metrics about feature correlation, sample-sample correlation, library size, etc. But, what about spatial correlation (e.g., Moran's I or similar). Perhaps comparing the distribution of Moran's I across genes in a simulated and real dataset would be a good first start.

We would like to clarify that synthspot does not actually simulate the spatial location of spots, but synthetic regions where spots from the same region share similar compositions. Hence, incorporating a spatial metric in the comparison is not feasible. However, as RCTD is the only method that explicitly uses spot locations in its model (Supplementary Table 2, "Location information"), we believe that generating synthetic datasets with actual coordinates would not significantly impact the conclusions of the study.

**Reviewer #2 (Public Review)**
On the other hand, the authors state that in silver standard datasets one half of the scRNA-seq data was used for simulation and the other half was used as a reference for the algorithms, but the method of splitting the data, i.e., at random or proportionally by cell type, was not specified.

The data was split proportionally by cell type. To clarify this, we have included an additional sentence in the main text under the first paragraph of “Cell2location and RCTD perform well in synthetic data”, as well as in Figure S2.

**Reviewer #2 (Recommendation for the authors)**
Figure legends in Figures 3, 4 and across most Supplementary material are almost illegible. Please consider increasing font size for better readability.

Thank you for bringing this to our attention. The font size has been increased for all main and supplementary figures. Additionally, the supplementary figures have also been exported in higher resolution.

Supplementary Notes Figure 2c reads "... total count per sampled multiplied by..."

This has been adapted, as well as the captions of Supplementary Notes Figure 3c and 4c which had the same typo.

**Review #3 (Public review)**
The simulation setup has a significant weakness in the selection of reference single-cell RNAseq datasets used for generating synthetic spots. It is unclear why a mix of mouse and human scRNA-seq datasets were chosen, as this does not reflect a realistic biological scenario. This could call into question the findings of the "detecting rare cell types remains challenging even for top-performing methods" section of the paper, as the true "rare cell types" would not be as distinct as human skin cells in a mouse brain setting as simulated here.

We appreciate the reviewer’s concern and would like to clarify that within one simulated dataset, we never mix mouse and human scRNA-seq data together. The synthetic spots generated for the silver standards are always sampled from a single scRNA-seq or snRNA-seq dataset. Specifically, for each of the seven public scRNA-seq datasets, we generate synthetic datasets with one of nine abundance patterns, resulting in a total of 63 synthetic datasets. These abundance patterns only affect the sampling priors that are used—the spots are still created with combinations of cells sampled from the same dataset.

Furthermore, it is unclear why the authors developed Synthspot when other similar frameworks, such as SRTsim, exist. Have the authors explored other simulation frameworks?

While there are other simulation frameworks available now, synthspot was designed to specifically address the requirements of our study, offering unique capabilities that make it suitable for deconvolution evaluation. Moreover, many of the simulators did not exist when we initially developed our tool. We have added the following rationale for why we developed synthspot in “Synthspot allows simulation of artificial tissue patterns”:

“On the other hand, general-purpose simulators are typically more focused on other inference tasks, such as spatial clustering and cell-cell communication, and are unsuitable for deconvolution. For instance, generative models and kinetic models like those of scDesign3 and scMultiSim are computationally intensive and unable to model entire transcriptomes. SRTSim focuses on modeling gene expression trends and does not explicitly model tissue composition, while spaSim only models tissue composition without gene expression.”

In our response to Reviewer 1 copied below, we also outline specific reasons why other simulators were not suitable for our benchmark:

• scDesign3: The runtime of scDesign3 is quite long as a result of its generative model. The example provided in its tutorial only simulates 183 genes and takes over seven minutes when using four cores on a system with Intel Xeon E5-2640 CPUs running at 2.5GHz. In a small downsampling analysis, we simulated 10, 50, 100, and 150 genes with scDesign3 and observed runtimes of 30, 130, 245, and 360 seconds, respectively. This seems to indicate a linear relationship between the number of genes and the runtime, therefore rendering it unsuitable for simulating whole-transcriptome datasets for deconvolution.

• spaSim: spaSim focuses on modelling cell locations in different tissue structures but does not provide gene expression data. It is designed for testing cell colocalization capabilities rather than simulating gene expression.

• SPIDER: Although SPIDER appears to have some overlap with our work, it seems to be in the early stages of development. The GitHub repository contains only two scripts without any documentation, and the preprint does not provide instructions on how to use the tool.

• SRTSim: SRTSim explicitly states in its publication that it is not suitable for evaluating cell type deconvolution, as its focus is on simulating gene expression data without modelling cell type composition.

• scMultiSim: scMultiSim, like scDesign3, is limited in its capability to model the entire transcriptome.

Finally, we would have appreciated the inclusion of tissue samples with more complex structures, such as those from tumors, where there may be more intricate mixing between cell types and spot types.

We acknowledge the reviewer's suggestion and have incorporated a melanoma dataset from Karras et al. (2022) in response to this suggestion. This study profiled melanoma tumors by using both scRNA-seq and spatial technologies. The scRNA-seq consists of eight immune cell types and seven melanoma cell states. We have included this study as an additional silver standard and case study, the latter of which is presented in a separate section following the liver analysis (and a corresponding section in Methods).

We found that method performances on synthetic datasets generated from this melanoma dataset follow previous trends (Figure S3-S5). However, the inclusion of the case study led to the following changes in the overall rankings: cell2location and RCTD are now tied for first place (previously RCTD ranked first), and Seurat and SPOTlight have swapped places. Despite these changes, the core messages and conclusions of our paper remain unchanged. All relevant figures (Figures 1a, 2, 3a, 4a, 6b, 7a, S3-S6, S9) have been updated to incorporate these new analyses and results.

**Review #3 (Recommendation for the authors)**
To maintain consistency in the results, it is recommended to exclude the human scRNAseq set when generating synthetic spots. Furthermore, addressing the other significant weaknesses mentioned earlier would be beneficial.

Please refer to our response to the public review where we address the same remark.

It is essential to differentiate this work from previous benchmarking and simulation frameworks.

In addition to the rationale on why we developed our own framework (see response to the public review), we have included the following text in the discussion that highlights our versatile approach when using a real spatial dataset for evaluation:

“In the case studies, we demonstrated two approaches for evaluating deconvolution methods in datasets without an absolute ground truth. These approaches include using proportions derived from another sequencing or spatial technology as a proxy, and leveraging spot annotations, e.g., zonation or blood vessel annotations, that typically have already been generated for a separate analysis.”

Furthermore, we conducted an extra analysis in the liver case study, generating synthetic datasets with one experimental protocol and using the remaining two as separate references (Figure S13). This further illustrates the usefulness of our simulation framework, which we mentioned by appending this sentence in the discussion:

“As in our silver standards, users can select the abundance pattern most resembling the real tissue to generate the synthetic spatial dataset, as we have also demonstrated in the liver case study.”